# ADVERSARIAL MACHINE LEARNING IN LATENT REPRESENTATIONS OF NEURAL NETWORKS

## ABSTRACT

Distributed deep neural networks (DNNs) have been shown to reduce the computational burden of mobile devices and decrease the end-to-end inference latency in edge computing scenarios. While distributed DNNs have been studied, to the best of our knowledge the resilience of distributed DNNs to adversarial action remains an open problem. In this paper, we fill the existing research gap by rigorously analyzing the robustness of distributed DNNs against adversarial action. We cast this problem in the context of information theory and introduce two new measurements for distortion and robustness. Our theoretical findings indicate that (i) assuming the same level of information distortion, latent features are always more robust than input representations; and (ii) the adversarial robustness is jointly determined by the DNN feature dimension and the generalization capability. To test our theoretical findings, we perform extensive experimental analysis by considering 6 different DNN architectures, 6 different approaches for distributed DNN and 10 different adversarial attacks to the ImageNet-1K dataset. Our experimental results support our theoretical findings by showing that the compressed latent representations can reduce the success rate of adversarial attacks by 88% in the best case and by 57% on the average compared to attacks to the input space.

## 1 INTRODUCTION

Deep neural networks (DNNs) have achieved significant success in various domains such as computer vision (Kirillov et al., 2023), natural language processing (OpenAI, 2023), and wireless communication (Baldesi et al., 2022), among many others. However, state-of-the-art DNNs are challenging to deploy on resource-limited mobile devices. While mobile-specific DNNs have been proposed (Sandler et al., 2018), they usually come with a significant loss in accuracy. On the other hand, completely offloading the computation to edge or cloud computers is infeasible in mobile scenarios due to the excessive communication overhead corresponding to the transfer of the DNN input from the mobile device to the edge/cloud (Wang et al., 2019a). A new paradigm called *distributed computing* – also referred to as *split computing* in prior art – divides the computation of DNNs across multiple devices – according to the available processing power and networking bandwidth. The key advantage is that optimal load distribution can be achieved while meeting maximum end-to-end latency constraints and also preserving the DNN accuracy (Matsubara et al., 2022a). For an excellent survey on distributed/split computing, the reader is referred to the work by Matsubara et al. (2021).

Although prior work has proven the advantages of distributing the DNN computation, it is also evident that this approach opens the door to adversarial attacks to intermediate (latent) representations. Figure 1 shows a high-level overview of the adversarial scenario under consideration. Without loss of generality, we assume that a DNN model is divided into a *mobile DNN* and a *local DNN*, respectively executed by the mobile device and an edge/cloud computer. Usually, the DNN architecture is modified by introducing a compression layer at the end of the mobile DNN (Eshratifar et al., 2019b; Matsubara et al., 2019; Hu & Krishnamachari, 2020; Shao & Zhang, 2020; Matsubara et al., 2020), which is trained to learn a *latent representation* that reduces the amount of data being sent to the edge/cloud. This way, the output tensor of the mobile DNN is transmitted to the edge/cloud server instead of the input data. The compressed representation is then used by the local DNN to produce the final prediction output (*e.g.*, classification). The distributed nature of the computation exposes the latent representation to adversarial action. Indeed, due to the need to communicate the latent representation across devices over a wireless network, an adversary can easily eavesdrop the latent representation and craft an adversarial sample to compromise the local DNN as shown in Figure 1.

Despite its significance and timeliness, to the best of our knowledge, assessing the robustness of distributed DNNs remains an unexplored problem. We remark that achieving a fundamental understanding of these attacks and evaluating their effectiveness in state-of-the-art DNNs is paramount to design robust distributed DNNs. To this end, we theoretically analyze the robustness of distributed DNNs using information theory – specifically, we build on notions from Information Bottleneck (IB) theory by Tishby et al. (2000) and propose two new measurements for distortion and robustness that are general to all DNN models and can be leveraged to analyze them. Our first key theoretical finding is that with similar levels of information distortion, *latent representations are always more robust than input representations*. In other words, *distributed DNNs are intrinsically a better solution for distributed/mobile computing systems than traditional DNNs*. Our second key finding is that *the DNN robustness is intrinsically related to the cardinality of the latent space*. Intuitively, this is because the search space available to the attacker is smaller. On the other hand, while a smaller latent space may increase robustness by reducing the model variance, it will also introduce bias in the model thus affecting the generalization capability of the DNN model.

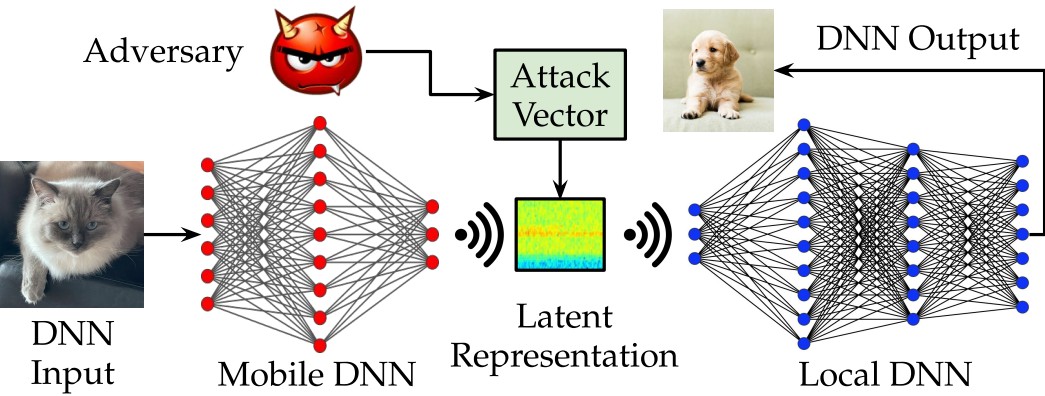

Figure 1: Overview of Adversarial Attacks to Distributed DNNs.

We extensively evaluate our theoretical findings by considering 10 adversarial algorithms, i.e., 4 white-box attacks (Goodfellow et al., 2014; Kurakin et al., 2018; Dong et al., 2018; Madry et al., 2017) and 6 black-box attacks (Ilyas et al., 2018; Li et al., 2019; Andriushchenko et al., 2020; Dong et al., 2019; Cheng et al., 2019; Wang et al., 2022). We apply these attacks to 6 reference architectures (Simonyan & Zisserman, 2014; He et al., 2016) designed with 6 distributed DNN approaches (Eshratifar et al., 2019a; Shao & Zhang, 2020; Matsubara et al., 2020; 2022a; Singh et al., 2020; Matsubara et al., 2022c). The experimental results validate our theoretical findings on the examined DNNs and attack algorithms.

The key contributions of this paper can be summarized as follows:

• To the best of our knowledge, we are the first to investigate the robustness of distributed DNNs against adversarial action. We leverage notions of IB theory and propose two new metrics for distortion and robustness of distributed DNNs. We theoretically prove that distributed DNNs are less vulnerable to perturbations of similar magnitude comparing to traditional DNNs, and that a latent representation with lower dimensions enhances robustness by reducing the DNN variance;

• We perform extensive experiments with the ImageNet-1K (Deng et al., 2009) dataset, by considering 6 different DNN architectures, 6 different distributed DNN approaches under 10 different attacks to support our theoretical findings. The results show that the theoretical analysis applies to the experimental settings under consideration. More precisely, the success rates of attacking the inputs is up to 88% higher than attacking latent representations (57% on average). We share our code for reproducibility at `https://github.com/asdfqwezxcf/AdvLatent`, and we hope that this work may open the door to a new field dedicated to studying the resilience of distributed DNNs.

This paper is organized as follows. Section 2 summarizes the related work on distributed DNNs and adversarial attacks to DNNs. Next, Section 3 presents our theoretical analysis based on IB. Section 4 discusses our experimental setup while Section 5 presents our experimental results. Finally, Section 6 draws conclusions and discusses possible directions for future work.

## 2   RELATED WORK

**Distributed Neural Networks.** There is an increasing need for DNNs-based computation on mobile devices. Lightweight DNNs specifically tailored for mobile devices (Sandler et al., 2018; Tan et al., 2019; Howard et al., 2019) fail to achieve comparable performance with the state-of-the-art DNNs. While edge computing approaches maintain similar performance, they incur in excessive latency (Yao et al., 2020). As an intermediate option, Kang et al. (2017) divide a DNN into two parts executed at the mobile device and edge, respectively. However, such division leads to excessive networking load due to large latent space of DNNs. Other work has addressed this problem by introducing a "bottleneck" layer before the division point (Eshratifar et al., 2019b; Shao & Zhang, 2020; Eshratifar et al., 2019a). However, naive bottlenecks suffer from noticeable task-specific performance loss. Recent work utilizes more advanced training techniques such as knowledge distillation to preserve accuracy while achieving high in-network compression ratio (Matsubara et al., 2020).

Different from unsupervised methods where compressed representations are learned for reconstruction purposes (Yang et al., 2023), supervised compression techniques aim to extract compact features relevant to the downstream task (Singh et al., 2020). However, such studies mainly aim to optimize the rate-distortion metric while often neglect the limited computational capability of mobile devices by introducing bottlenecks in the last layers (Ballé et al., 2018; Minnen et al., 2018; Datta et al., 2022; Ahuja et al., 2023). Inspired by ideas such as the reparameterization trick by Kingma & Welling (2013), and quantization with entropy coding by Ballé et al. (2016), Matsubara et al. (2022c) use stochastic bottleneck with learnable prior for entropy coding to optimize the three-way tradeoff between (a) minimizing the computational complexity of the mobile DNN, (b) minimizing the size the wirelessly transferred data, and (c) minimizing the DNN performance loss.

**Adversarial Machine Learning.** Adversarial attacks can be categorized as gradient-based, score-based, and decision-based. In *gradient-based scenarios*, attackers can obtain the input gradient through backpropagation and craft adversarial samples with gradient ascending. Fast Gradient Sign Method (FGSM) by Goodfellow et al. (2014) crafts adversarial samples in the $l_\infty$ space based on the one-step input gradient sign. Basic Iterative Method (BIM) by Kurakin et al. (2018) increases the effectiveness of FGSM by iteratively updating adversarial samples with multiple gradient steps. Momentum Iterative Method (MIM) by Dong et al. (2018) introduces momentum to iterative attacks which improves the transferability of adversarial samples. Projected Gradient Descent (PGD) by Madry et al. (2017) generalizes iterative attacks to $l_p$ space with a random start. Carlini & Wagner (2017b) form the attack as an optimization problem and evaluate different optimization algorithms with multiple loss functions. In black-box settings, gradient-based attacks find adversarial samples using gradients from a set of substitute DNNs. Recent work by Wang et al. (2021a) and Zhang et al. (2022b) improves the transferability of crafted adversarial examples with advanced gradient design.

Different from gradient-based approaches, *score-based adversaries* can only access the scores for every class given by the DNN. Natural Evolutionary Search (NES) by Ilyas et al. (2018) applies evolutionary algorithm to estimate gradient within limit queries. N-Attack by Li et al. (2019) designs a learnable Gaussian distribution centered around the input to generate random noise which makes a benign sample become an adversarial sample. Square Attack by Andriushchenko et al. (2020) adds a localized square-shaped perturbation at a random position to the original sample in each iteration. *Decision-based attacks* assume the adversary is only aware of the label having the highest score in the DNN output. Evolutionary Attack (EVO) by Dong et al. (2019) minimizes the distance with evolutionary search in the input space, while Hop-Skip-Jump Attack (HSJA) by Chen et al. (2020) designs a zeroth-order optimization algorithm to find minimum-magnitude perturbations with binary search. Sign-OPT Attack (S-OPT) by Cheng et al. (2019) accelerates the convergence by estimating the gradient with the sign of the directional derivative, while Triangle Attack by Wang et al. (2022) minimizes the adversarial perturbation in a smaller frequency space with the geometric property.

**Evaluating the Robustness of Neural Networks.** Probably Approximately Correct (PAC) learning has been used to analyze the adversarial robustness of DNNs (Montasser et al., 2019; Bhattacharjee et al., 2023; Attias et al., 2019; Awasthi et al., 2019; Bubeck & Sellke, 2021; Ashtiani et al., 2023). However, such work mainly attempts to find a lower bound for the dataset size which can attain a desired robustness level (Montasser et al., 2019; Attias et al., 2019; Ashtiani et al., 2023; Bhattacharjee et al., 2023). Conversely, our investigation is aimed at evaluating adversarial robustness in DNNs that are trained with certain datasets. In addition, these approaches are evaluated on simple DNNs, e.g., 2-layer neural networks (Awasthi et al., 2019; Bubeck & Sellke, 2021), while we eval-

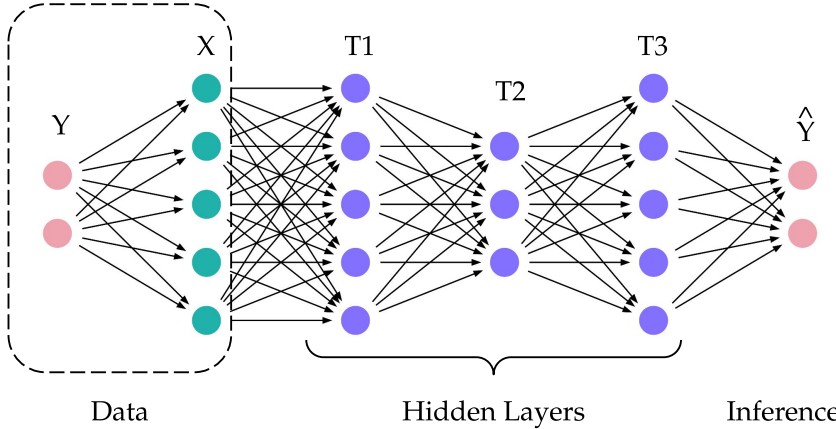

Figure 2: Modeling DNN with IB. Each representation $T_i$ only depends on the previous output $T_{i-1}$, and the optimal $T_i^*$ can be interpreted as the IB solution which optimizes Equation 1 at layer $i$.

uate our findings on state-of-the-art DNNs. Carlini et al. (2019) propose a set of criteria to evaluate adversarial robustness with numerical results, while Carlini & Wagner (2017a); Dong et al. (2020); Croce et al. (2020) evaluate the robustness of different defense approaches. The key issue is that different work comes to contradictory conclusions. For example, Su et al. (2018) argue that there is a tradeoff between generalization and robustness while Stutz et al. (2019) state that generalization does not affect the robustness. Tsipras et al. (2018) argue robust training sacrifice accuracy on standard datasets while Ilyas et al. (2019) can achieve same accuracy on both adversarial and standard dataset. Tishby & Zaslavsky (2015) first propose to use IB (Tishby et al., 2000) to analyze DNNs and Shwartz-Ziv & Tishby (2017); Saxe et al. (2019) analyze the generalization and compression capability of DNNs with experiments. However, such studies heavily rely on the variational approximation which requires increasing number of samples with respect to their dimension to reduce the bound on estimation error (Poole et al., 2019). Thus, the above-mentioned works can only perform experiments on relatively small neural networks. In stark opposition with this paper, recent work on IB for robust learning (Alemi et al., 2016; Amjad & Geiger, 2019; Wang et al., 2021b; Kim et al., 2021) does not provide a general analysis for robustness of DNNs.

## 3   ROBUSTNESS ANALYSIS OF DISTRIBUTED DEEP NEURAL NETWORKS

**3.1: Background on Information Bottleneck (IB).** The IB is a model-agnostic information-theoretical framework introduced by Tishby et al. (2000) to extract the relevant information about a random variable (r.v.) $Y$ from another r.v. $X$ by finding a representation $T$ which compresses the information of $X$ while captures only the sufficient information about $Y$. As shown in Figure 2, we model a DNN with a Markov chain $Y \mapsto X \mapsto T_1 \mapsto \cdots \mapsto T_k \mapsto \hat{Y}$, where $X, Y, \hat{Y}$ and $T_i$ are respectively the input, its label, the inference output and the output of the $i$-th hidden DNN layer. The IB optimizes the following:

$$\min_{P(T_i|X)} I(X; T_i) - \beta \cdot I(Y; T_i), 1 \leq i \leq k \tag{1}$$

where $I(X; T_i)$ is the mutual information between $X$ and $T_i$ while $I(Y; T_i)$ is the mutual information between $Y$ and $T_i$. Each layer can be thus described by its own unique information plane $(I(X; T_i), I(Y; T_i))$ which represents its compression and generalization capability. Notice that optimizing Equation 1 is equivalent to minimizing $I(X; T_i)$ – i.e., learning to compress – whereas maximizing $I(Y; T_i)$ – i.e., learning to generalize. To simplify notation, and without loss of generality, henceforth we will consider a single generic hidden layer $T$.

**3.2: Variance vs Bias in Adversarial Attacks.** We define the end-to-end robustness of the DNN model as $I(Y; \hat{Y})$, which measures the mutual information between the input label (i.e., ground truth) and the DNN inference. We apply the Data Processing Inequality (DPI) to describe the information loss during processing (Cover, 1999):

$$I(Y; X) \geq I(Y; T) \geq I(Y; \hat{Y}) \tag{2}$$

In short, the generalization metric $I(Y; T)$ of hidden layer $T$ also describes the upper bound of $I(Y; \hat{Y})$, which is intrinsically a measure of robustness at layer $T$. By assuming that adversarial perturbations are not observable, it follows there is a prior yet unknown optimal solution $I^*(Y; T)$

for a specific DNN architecture that satisfies the IB where adversarial perturbations cannot decrease the performance – in other words, $I^*(Y;T)$ is resilient to adversarial attacks. The key issue is that although each DNN has a hypothesis set defined by its parameters, the optimum parameter set exhibiting the largest $I^*(Y;T)$ is unknown. To this end, each trained DNN using a finite dataset $(X, Y)$ has its own estimation $I(Y;T)$. Shamir et al. (2010) have proven that the estimated mutual information using finite samples has the following error bound:

$$||I^*(Y;T) - I(Y;T)|| \leq \mathcal{O}(\frac{|\mathcal{T}||\mathcal{Y}|}{\sqrt{n}}), \tag{3}$$

where $n$ denotes the number of data and $|\mathcal{T}|$, $|\mathcal{Y}|$ are the cardinality of $T$ and $Y$, respectively. Equation 3 is the information version of the complexity-generalization tradeoff in PAC learning. A larger latent space $|\mathcal{T}|$ (i.e., a more complex hypothesis set in PAC learning) will have larger variance resulting in decreased performance with inputs coming from a distribution different than $X$, which is described by the upper bound $||I^*(Y;T) - I(Y;T)||$. Conversely, with a smaller latent space (i.e., a smaller hypothesis set), the DNN has more bias, which leads to less accuracy. Equation 3 is also in line with Simon-Gabriel et al. (2019), which states that robustness of DNNs decreased with growing data dimension. The following holds.

> **Key Theoretical Finding #1: Variance vs Bias in Adversarial Attacks**
>
> For adversarial attacks to a hidden layer $T_{adv}$, the performance $I(Y;T_{adv}) \leq I^*(Y;T) - \mathcal{O}(|\mathcal{T}||\mathcal{Y}|/\sqrt{n})$ is jointly determined by $I^*(Y;T)$ and $\mathcal{O}(|\mathcal{T}||\mathcal{Y}|/\sqrt{n})$. In other words, *in distributed DNNs, the feature compression layer helps to enhance the adversarial robustness by reducing the variance but also introduce vulnerability as a result of adding bias.*

**3.3: Attacks in Latent Space vs Input Space.** A key challenge to compare adversarial action in distributed DNNs versus conventional DNNs is that the input space and latent representation space have different cardinality. As such, we utilize a new metric based on the Kullback-Leibler (KL) divergence $D_{KL}[P(Y|X)||P(Y|T)]$ to describe the information distortion (Tishby & Zaslavsky, 2015). Since $D_{KL}$ is a function of random variables $X$ and $T$, the expectation of $D_{KL}$ is

$$
\begin{aligned}
\mathbb{E}\{D_{KL}\} &= \sum_{X,T} P(X,T) \sum_Y P(Y|X,T) \log \frac{P(Y|X)}{P(Y|T)} \\
&= \sum P(X,T,Y) \log \frac{P(Y|X)P(X|T)}{P(Y|T)P(X|T)} \\
&= \sum P(X,T,Y) \log \frac{P(X,Y|T)}{P(Y|T)P(X|T)} \\
&= I(X;Y|T)
\end{aligned}
\tag{4}
$$

The conditional mutual information $I(X;Y|T)$ can be considered as the *residual information* between $X$ and $Y$ which is not captured by $T$. Due to the chain rule of mutual information,

$$I(X;Y|T) = I(X,T;Y) - I(Y;T) \tag{5}$$

For a Markov chain $Y \mapsto X \mapsto T$, the joint distribution $P(X,Y,T)$ has following property

$$P(X,Y,T) = P(T|X,Y)P(Y|X)P(X) = P(T|X)P(Y|X)P(X) \tag{6}$$

Therefore, $I(X,T;Y)$ can be simplified as

$$
\begin{aligned}
I(X,T;Y) &= \mathbb{E}\left\{\log \frac{P(X,T,Y)}{P(X,T)P(Y)}\right\} = \mathbb{E}\left\{\log \frac{P(T|X)P(Y|X)P(X)}{P(T|X)P(X)P(Y)}\right\} \\
&= \mathbb{E}\left\{\log \frac{P(Y|X)}{P(Y)}\right\} = I(X;Y)
\end{aligned}
\tag{7}
$$

From Equation 5 and Equation 7, it follows that

$$I(X;Y|T) = I(X;Y) - I(Y;T). \tag{8}$$

For adversarial attacks in conventional DNNs, the adversarial samples are generated in the input space. In this case, the Markov chain is $Y \mapsto X_{adv} \mapsto T'$, where $X_{adv}$ represents adversarial samples and $T'$ is the corresponding latent representation. Similarly, the Markov chain for adversarial

attacks in distributed DNNs is $Y \mapsto X \mapsto T_{adv}$ where $T_{adv}$ represents adversarial latent samples. Intuitively, introducing adversarial perturbations will confuse DNNs, thus increasing the *residual information*. As such, we can use the residual information to quantify the adversarial perturbations. Assuming $X_{adv}$ and $T_{adv}$ have same level of information distortion,

$$I(X_{adv}; Y) - I(Y; T') = I(X; Y) - I(Y; T_{adv}). \tag{9}$$

Since $X_{adv}$ is a mapping of $X$, there is a Markov chain $Y \mapsto X \mapsto X_{adv}$. The following holds.

---

**Key Theoretical Finding #2: Attacks in Latent Space vs Input Space**

By DPI, it follows that $I(X; Y) \geq I(X_{adv}; Y)$. Therefore, it follows that

$$I(Y; T') \leq I(Y; T_{adv}). \tag{10}$$

In other words, *with same level of information distortion, attacking the latent space is less effective than attacking the input space.*

---

## 4   EXPERIMENTAL SETUP

### 4.1   ATTACKS UNDER CONSIDERATION

We have extensively validated the theoretical findings obtained in Section 3 by implementing 10 popular attacks to DNNs. These include 4 gradient-based white-box attacks by Goodfellow et al. (2014), Kurakin et al. (2018), Dong et al. (2018) and Madry et al. (2017), as well as 3 score-based black-box attacks (Ilyas et al., 2018; Li et al., 2019; Andriushchenko et al., 2020) and 3 decision-based black-box attacks (Dong et al., 2019; Cheng et al., 2019; Wang et al., 2022). We first formally define adversarial attacks in input and latent space, and then describe the related algorithms.

**Adversarial Attacks in Input Space.** Let $f : \mathbb{R}^d \to \mathbb{C}^k$ denote a DNN where $\mathbb{R}$ and $\mathbb{C}$ are respectively the input and output space, and $d$ and $k$ are the corresponding dimension of these two spaces. The DNN will assign highest score to the correct class $y = \arg\max_k f(x)$ for each input $x$. The adversarial goal is to introduce a perturbation $\delta_d \in \mathbb{R}^d$ to the original sample so that

$$\arg\max_{k=1,...,K} f(x + \delta_d) \neq y, \tag{11}$$

where $||\delta_d||_p \leq \sigma$ and $\sigma$ is the distance constraint under different $l_p$ norm. Additionally, for visual applications, $\delta_d$ should satisfy the condition $x + \delta_d \in [0, 1]^d$ as there is an explicit upper and lower bound for red, green and blue (RGB) value in digital images.

**Adversarial Attacks in Latent Space.** Let $g : \mathbb{R}^d \to \mathbb{H}^t$ and $f : \mathbb{H}^t \to \mathbb{C}^k$ denote the mobile DNN and local DNN, where $\mathbb{H}$ and $t$ are the latent space and its associated dimension, respectively. For each input $x$, the mobile DNN will generate a corresponding latent representation $g(x) \in \mathbb{H}^t$ and the local DNN will generate output $y = arg\max_k f(g(x))$ by taking the latent representation as input. Adversarial action in latent space adds a perturbation $\delta_t \in \mathbb{H}^t$ such that

$$\arg\max_{k=1,...,K} f(g(x) + \delta_t) \neq y, \tag{12}$$

where $||\delta_t||_p \leq \sigma$ is the distance constraint under $l_p$ norm. We remark that the latent representations are model-dependent and there is no explicit bound for their value other than their computer-level representation (e.g., float, integer, double).

**White-box Attacks:** We consider 4 gradient-based attacks only in white-box setting because latent representations are different for each model, resulting in numerous surrogate DNNs that may be infeasible in practical settings. We choose FGSM, BIM, MIM with $l_\infty$ norm constraints. PGD is implemented for both $l_2$ and $l_\infty$ spaces as a baseline for other black-box attacks.

**Black-box Attacks:** We consider 3 score-based attacks NES, N-Attack and Square Attack in $l_\infty$ space and 3 decision-based attacks EVO, S-OPT and Triangle Attack in $l_2$ space. During our experiments, we found that S-OPT and HSJA have similar results, so we do not report HSJA due to space limitations.

**Dataset and Metrics:** We evaluate adversarial robustness using 1000 samples from the validation set of ImageNet-1K (Deng et al., 2009), limiting the samples to those which are correctly classified.

Table 1: List of feature compression approaches considered in this paper

| Category | Approach | Description |
|---|---|---|
| Dimension | SC | naive supervised compression trained with cross entropy |
| | KD | bottleneck trained with naive knowledge distillation |
| | BF | multi-stage training with distillation and cross entropy |
| Data | JC | reduce precision in frequency domain using JPEG approach |
| | QT | uniformly compress every element using naive bit quantization |
| Advanced | ES | bottleneck trained with distillation and information-based loss and data compressed with quantization and entropy coding |

We define the perturbation budget $\epsilon$ as the mean square error (MSE) under the $l_2$ norm constraint (i.e., $\epsilon \times d = \sigma^2$ and $\epsilon \times t = \sigma^2$ in input and latent space respectively.) and the maximum element-wise distance under $l_\infty$ norm constraint (i.e., $\epsilon = \sigma$), we define the attack success rate (ASR) as

$$\text{ASR}(\epsilon) = \frac{1}{N} \sum_{i=1}^{N} \mathbf{I} \left\{ \arg\max_{k=1,\dots,K} f(x_i, \delta_i) \neq y_i \right\}, \tag{13}$$

where $\mathbf{I}\{\cdot\}$ is the indicator function and $f(x_i, \delta_i)$ is the DNN output when fed with the $i$-th sample.

## 4.2 Deep Neural Networks Under Consideration

**DNN Architectures.** First, we consider 3 DNNs: VGG16 from Simonyan & Zisserman (2014) as well as Resnet50 and Resnet152 from He et al. (2016). Next, to investigate the effect introduced by the feature compression layer (i.e., the "bottleneck") proposed for distributed DNNs, we introduce the same bottleneck design as Matsubara et al. (2022b) to VGG16, Resnet50 and Resnet152 and denote the new architectures as VGG16-fc, Resnet50-fc and Resnet152-fc.

**Different Compression Approaches.** In distributed DNNs, compression can be achieved both by *compressing the dimensionality* with bottlenecks and *compressing the data size* with coding and precision reduction. We consider 3 different bottleneck training strategies for *dimension reduction*: Supervised Compression (SC) (Eshratifar et al., 2019b; Shao & Zhang, 2020), Knowledge Distillation (KD) (Matsubara et al., 2019) and BottleFit (BF) (Matsubara et al., 2022a). We also choose 2 *data compression* strategies JPEG Compression (JC) (Alvar & Bajić, 2021), Quantization (QT) (Singh et al., 2020) as well as 1 *advanced* approach Entropic Student (ES) that both compress the dimension and data size (Matsubara et al., 2022c). We summarize these approaches in Table 1.

## 5 Experimental Results

### 5.1 Performance With Different DNN Architectures

Figure 3 shows the ASR obtained on ResNet152-fc with perturbation budget $\epsilon = 0.01$ (we explore the performance as a function of the perturbation budget in Figure 6). Remarkably, we notice that the ASR is higher for attacks in the input space than attacks in the latent space for each attack algorithm considered. In the case of Triangle Attack, the latent ASR is 88% less than the input ASR. On average, the ASR in input is 57.49% higher than the ASR obtained by attacks in the latent space. Moreover, Square Attack, EVO, and Triangle Attack have lowest ASR on latent representations. This is because these attacks search perturbations in a lower dimensional space, and hence it is more challenging for the adversary to find the effective distortions in compressed latent space. Figure 3 shows that our theoretical findings are general and apply to a wide variety of attacks. For this reason, due to space limitations, in the next experiments we only show results of one score-based attack in $l_\infty$ norm and one decision-based attack in $l_2$ norm as well as corresponding white-box baselines.

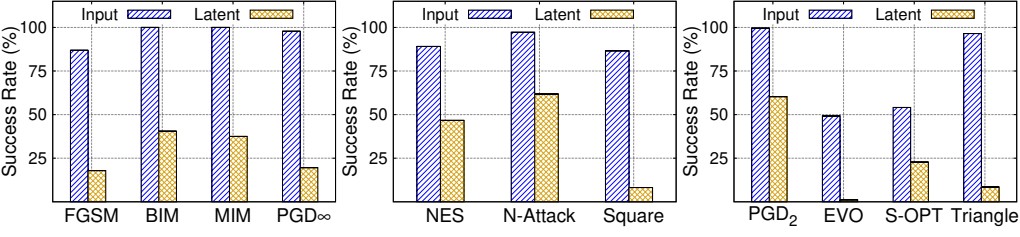

Figure 3: 10 different attacks to ResNet152-fc with perturbation budget $\epsilon = 0.01$.

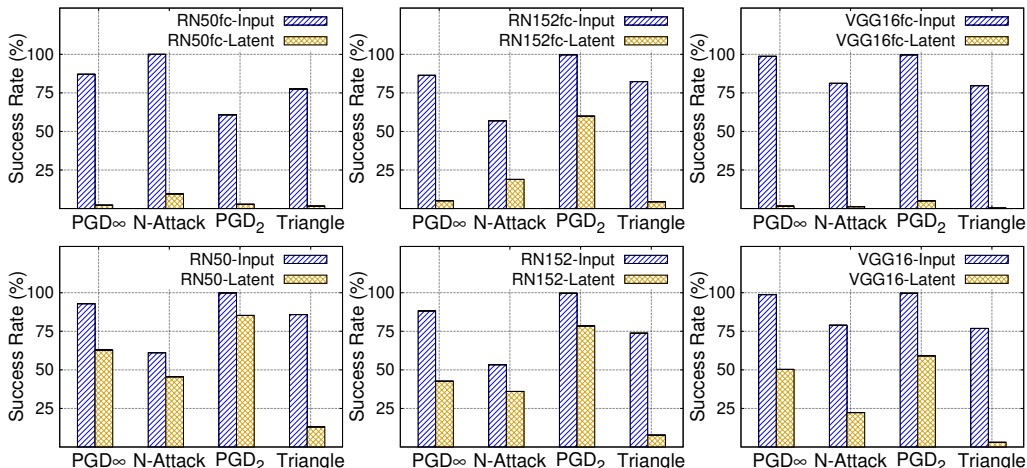

Figure 4: Whitebox baseline (PGD) and blackbox attacks under $l_\infty$ and $l_2$ in input and latent space with perturbation budget $\epsilon = 0.003$ applied to 6 different DNNs.

Figure 4 shows the performance of PGD, N-Attack and Triangle Attack on different DNNs with perturbation budget $\epsilon = 0.003$. For each DNN, the ASR is higher in input-space attacks. In VGG16-bf, which shows the best robustness, the average ASR in the latent space is 87.8% lower than input attacks. On average, latent representations are 58.33% more robust.

## 5.2 PERFORMANCE WITH DIFFERENT COMPRESSION APPROACHES

To evaluate the robustness of different compression approaches, we choose Square Attack and Triangle Attack which are the newest approaches for score-based and decision-based attacks respectively. We do not consider gradient-based attacks as compression approaches such as knowledge distillation add penalty terms to their loss function, which leads to gradient masking. Hence, their robustness cannot be correctly evaluated by naive gradient-based attacks (Athalye et al., 2018). We choose a larger perturbation budget ($\epsilon = 0.05$) than the experiments depicted in Figure 3 to further evaluate whether the compressed feature space is robust to attacks relying on low-dimensional sub-space searching. We note that data compression can be applied in addition to bottlenecks. However, for comparison purposes, we choose ResNet50 without bottlenecks for JC and QT.

Figure 5 shows the ASR of Square Attack in $l_\infty$ space and Triangle Attack in $l_2$ space with perturbation budget $\epsilon = 0.05$. Except the JC and QT, the adversarial robustness shows the same trend regardless of the examined approaches. The average ASRs in input space are 79.07% and 87.22% higher than the average ASRs in latent space for Square Attack and Triangle Attack respectively. For DNNs with bottlenecks, despite the increase in perturbation, the ASR of Square Attack and Triangle Attack performed in latent representations do not increase distinctively comparing to Figure 3. However, since JC and QT do not have separate feature compression layers, the ASR of Square Attack and Triangle Attack in input space are only 55.8%, 0.25% higher than the attacks in latent space, showing a significant downgrade comparing to the other 4 approaches. *These results confirm that the compressed feature space is indeed robust to attacks that search in lower dimensions.*

## 5.3 PERFORMANCE AS FUNCTION OF COMPRESSION RATIO

In the previous section, we have shown that the robustness of the latent representation is mostly characterized by the bottleneck layer properties rather than the compression approach itself. Thus, we further evaluate the robustness for different sizes of latent space using N-Attack, MIM under $l_\infty$ constraint and S-OPT, PGD under $l_2$ constraint with multiple perturbation budgets ($\epsilon = 0.003$; $\epsilon = 0.01$; $\epsilon = 0.03$). The cardinality of the latent space is controlled by the number of channels at the bottleneck layer. We first set the channel number as 12 for ResNet152-fc that can achieve 77.47% validation accuracy, which is almost similar to the performance of the original ResNet152 (78.31%). Then, we reduce the number of channels to 3, which decreases the dimension of latent representations but also reduces the end-to-end performance to 70.01%. We do not repeat the results for Square Attack and Triangle Attack since they fail to achieve satisfactory ASR in the previous experiments due to their smaller search subspace, as shown in Figures 3 and 5.

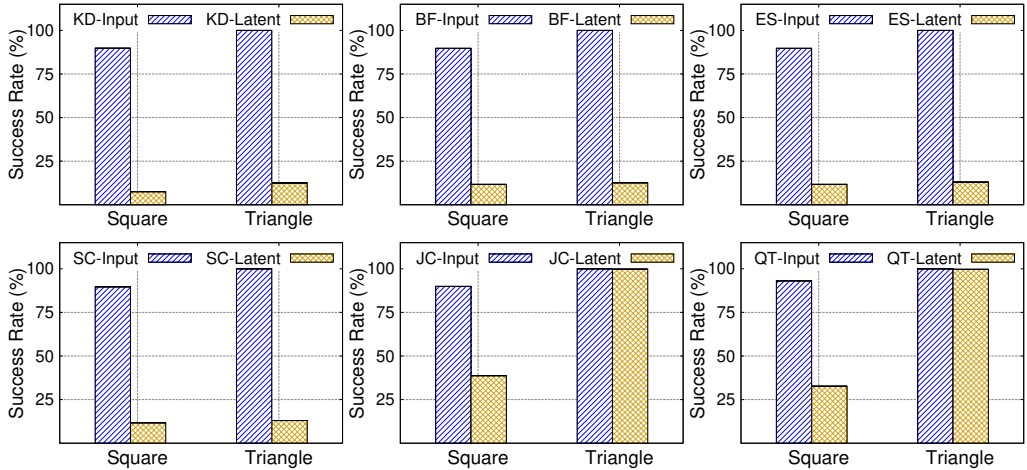

Figure 5: Square and Triangle attack success rate associated with 6 feature compression approaches with perturbation budget $\epsilon = 0.05$.

Figure 6 shows results obtained by considering the $l_\infty$ and $l_2$ attacks with multiple perturbation budgets ($\epsilon = 0.003$; $\epsilon = 0.01$; $\epsilon = 0.03$) in the latent space of original ResNet-152, 12-channel ResNet152-fc and 3-channel ResNet152-fc. From ResNet152 to 12-channel ResNet152-fc, the ASR reduces as the dimensionality of latent representations decreases by 213.33% – in other words, $\mathcal{O}(|\mathcal{T}||\mathcal{Y}|/\sqrt{n})$ is 21.33 times smaller. However, after reducing the channel size to 3, the ASR does not decrease any further[1]. Conversely, distributed DNNs with a smaller channel size become more vulnerable to perturbations. This is because when reducing channels from 12 to 3 channels, the accuracy also decreases to 7.46%, which in turn lessens the end-to-end generalization capability (i.e., $I^*(Y; T)$). *This experiment supports our analysis that the robustness in latent representations of distributed DNN is jointly determined by the end-to-end performance and feature dimensions.*

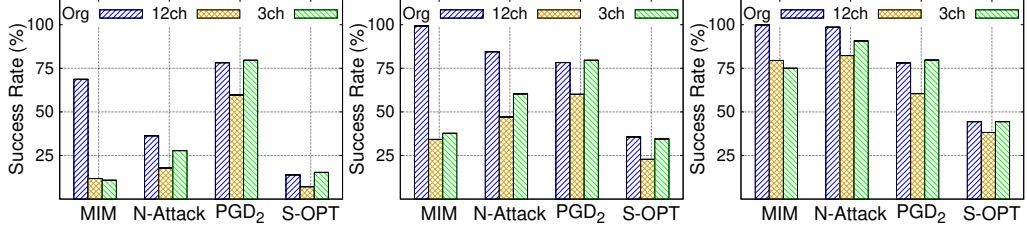

Figure 6: $l_\infty$ and $l_2$ attack success rate for different latent cardinalities of Resnet152-fc with different perturbation budgets: (left) $\epsilon = 0.003$; (center) $\epsilon = 0.01$; (right) $\epsilon = 0.03$.

## 6  CONCLUDING REMARKS

This paper has investigated adversarial attacks to latent representations of DNNs for the first time. First, we have theoretically analyzed the robustness of latent representations with information theoretical notions and based on the information bottleneck theory. To prove our theoretical findings, we have performed an extensive set of experiments with 6 different DNN architectures, 6 different distributed DNN approaches and considering 10 different attacks in literature. Our investigation concludes that latent representations are more robust than input representations assuming the same level of information distortion. Moreover, the adversarial robustness in latent space is jointly determined by the feature size and the end-to-end model generalization capability. Finally, we have shown that the success rate of attacks in the latent representations can be reduced by 88% in the best case and 57.49% on average compared to the same algorithms in input space. We hope that this work will inspire future work on the topic of adversarial machine learning on latent representations. We are currently working on designing defenses against attacks to latent representations of distributed DNNs investigated in this paper.

---

[1]Due to the difference of devices and random seeds, the ASR can vary 2-3%. Thus we do not consider the decrease of the MIM success rate in 3-channel ResNet152-fc which is less than 5%.

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

## A   DETAILS OF EXPERIMENTAL RESULTS

We provide all numerical results in Section 5 as follows. Table A-1 refers to the experiments in Figure 3 while Table A-2 refers to Figure 4. Table A-3 and Table A-4 refers to Figure 5 and Figure 6, respectively.

Table A-1: Different attacks to Resnet152-fc with perturbation budget $\epsilon = 0.01$. **Key findings:** (i) Latent representations are more robust regardless of attack algorithms; (ii) Attacks relying on low-dimension subspace search (Square Attack, EVO, Triangle Attack) achieve the lowest ASR in compressed latent space.

| Attacks | Input | Latent |
|---|---|---|
| FGSM | 86.9% | **17.9%** |
| BIM | 100% | **40.5%** |
| MIM | 100% | **37.5%** |
| PGD ($l_\infty$) | 97.8% | **19.6%** |
| NES | 89.1% | **46.7%** |
| N-Attack | 97.2% | **61.8%** |
| Square Attack | 86.6% | **8.2%** |
| PGD ($l_2$) | 99.6% | **60.3%** |
| EVO | 49.2% | **1.1%** |
| S-OPT | 54.1% | **22.8%** |
| Triangle Attack | 96.5% | **8.5%** |

Table A-2: Whitebox baseline (PGD) and blackbox attacks under $l_\infty$ and $l_2$ in input and latent space with perturbation budget $\epsilon = 0.003$ applied to 6 different DNNs. **Key findings:** (i) Latent representations are more robust for all DNN architectures; (ii) DNNs with bottlenecks achieve better robustness in latent representations.

| Models | PGD ($l_\infty$) | | N-Attack | | PGD ($l_2$) | | Triangle Attack | |
|---|---|---|---|---|---|---|---|---|
| | input | latent | input | latent | input | latent | input | latent |
| Resnet50 | 92.9% | **62.9%** | 61.1% | **45.5%** | 99.9% | **85.3%** | 85.8% | **13.0%** |
| Resnet152 | 88.2% | **42.7%** | 53.3% | **36.0%** | 99.7% | **78.5%** | 73.8% | **7.7%** |
| VGG16 | 98.8% | **50.3%** | 78.9% | **22.3%** | 99.7% | **59.1%** | 76.8% | **3.0%** |
| Resnet50-fc | 87.1% | **2.2%** | 100% | **9.5%** | 60.8% | **2.8%** | 77.5% | **1.6%** |
| Resnet152-fc | 86.4% | **5.0%** | 56.9% | **18.9%** | 99.7% | **59.9%** | 82.3% | **4.4%** |
| VGG16-fc | 98.7% | **1.6%** | 81.2% | **1.2%** | 99.7% | **5.0%** | 79.7% | **0.3%** |

Table A-3: Square and Triangle attack success rate associated with 6 feature compression approaches with perturbation budget $\epsilon = 0.05$. SC, KD, BF and ES share the same DNN with bottleneck while JC and QT are DNNs without bottleneck. **Key findings:** (i) DNNs with bottlenecks (SC, KD, BF, ES) achieve better robustness in latent representations than DNNs without bottlenecks (JC, QT) when attackers search on small subspace. (ii) ASR show the same trend for the same DNN architecture regardless of compression approaches. As such, robustness in latent representations are mainly characterized by the feature dimension.

| Compression Approaches | Square Attack | | Triangle Attack | |
|---|---|---|---|---|
| | input | latent | input | latent |
| SC | 89.7% | **11.7%** | 100% | **13.0%** |
| KD | 89.8% | **7.5%** | 100% | **12.5%** |
| BF | 89.7% | **11.7%** | 100% | **12.6%** |
| ES | 89.7% | **11.7%** | 100% | **13.0%** |
| JC | 89.9% | 38.7% | 100% | 99.8% |
| QT | 93.1% | 32.7% | 100% | 99.7% |

Table A-4: $l_\infty$ and $l_2$ attack success rate for different latent cardinalities of Resnet152-fc with different $\epsilon$. Accuracy without attacks: original Resnet152 (78.31%); 12-channel Resnet152-fc (77.47%); 3-channel Resnet152-fc (70.01%). **Key findings:** (i) Smaller feature cardinality improves the robustness when it affects little end-to-end performance (from 78.31% to 77.47%); (ii) Smaller feature cardinality reduces the robustness by degrading generalization capability (from 77.47% to 70.01%).

| Attacks | $\epsilon = 0.003$ | | | $\epsilon = 0.01$ | | | $\epsilon = 0.03$ | | |
| | org | 12ch | 3ch | org | 12ch | 3ch | org | 12ch | 3ch |
|---|---|---|---|---|---|---|---|---|---|
| MIM | 68.7% | 11.8% | **10.8%** | 99.3% | **34.2%** | 37.7% | 99.9% | 79.5% | **75.0%** |
| N-Attack | 36.3% | **17.8%** | 27.8% | 84.5% | **47.1%** | 60.3% | 98.6% | **82.3%** | 90.7% |
| PGD | 78.2% | **59.7%** | 79.6% | 78.3% | **60.1%** | 79.6% | 78.1% | **60.5%** | 79.7% |
| S-OPT | 13.9% | **7.10%** | 15.3% | 35.6% | **22.8%** | 34.5% | 44.4% | **38.2%** | 44.4% |

## B    DISTINCTION WITH OTHER WORK

**Adversarial Auto-encoders.** Makhzani et al. (2015) first proposed to apply adversarial training to auto-encoders. Since then, adversarial auto-encoders has become a popular approach for outlier detection (Pidhorskyi et al., 2018; Beggel et al., 2020; Salehi et al., 2021) and robust speech recognition (Espinoza-Cuadros et al., 2020; Latif et al., 2020; Sahu et al., 2018). The key idea of adversarial auto-encoder is to learn disentangled representations which is invariant to both clean and adversarial samples so that it can perform better outlier detection or speech recognition for unseen data. In this context, the latent representations are not considered to be exposed to attackers and the "robustness" of latent representations means *"semantically meaningful features"* for both clean and adversarial input.

On the other hand, our work consider attacks adding perturbations *directly to the latent space*. The scenario considered is extremely relevant to mobile scenarios since distributed DNNs have been proven to be extremely effective in reducing network load while preserving accuracy (Matsubara et al., 2022a;c; Shao & Zhang, 2020). In this case, latent representations are exposed to the attacker, which can lead to performance degradation when perturbed. Therefore, the "robustness" in latent representations means *"adversarially robust to attacks"*, which is conceptually different than adversarial auto-encoders.

**Latent-aware Adversarial Attacks.** After it first demonstrated that effective adversarial samples can be generated by manipulating the latent features (Sabour et al., 2015), latent-aware attacks became a new paradigm to evaluate the DNN robustness. For example, Yu et al. (2021) proposed a unified $l_\infty$ white-box attack to circumvent a wide range of defense strategies. Mopuri et al. (2017) proposed a universal attack independent to training data by fooling features in multiple layers. Wang et al. (2019b) proposed a physical attack algorithm for human de-identification by adding adversarial patterns which contain features of target class. Luo et al. (2022) crafted imperceptible samples based on semantic similarity in feature space. In black-box settings, latent similarity are generally used to improve the tranferability of adversarial samples (Huang et al., 2019; Inkawhich et al., 2019; 2020; Wang et al., 2021a).

However, our work separates itself from the above-mentioned literature since we focus on the adversarial perturbation of the latent representation itself, while latent-aware adversarial attacks use the latent representations as side information to craft adversarial samples in input space.

## C    AN ALTERNATIVE EXPLANATION OF KEY THEORETICAL FINDING #2

We clarify that the assumption of identical distortion in latent space is used to formally analyze the robustness of distributed DNNs and provide a fair comparison with attacks in the input space. On the other hand, we provide an alternative interpretation of our second theoretical finding that does not require the aforementioned assumption as follows.

Since we proposed to measure distortion with residual information $I(X; Y|T)$, the residual information of adversarial samples $I(X_{adv}; Y|T')$ will be larger than the residual information of benign samples $I(X; Y|T)$, i.e.,

$$I(X_{adv}; Y|T') \geq I(X; Y|T) \tag{14}$$

From Equation 8 in our paper, it follows that

$$I(X; Y) - I(X_{adv}; Y) \leq I(T; Y) - I(T'; Y) \tag{15}$$

In other words, the information loss between latent $T'$ and ground-truth $Y$ is larger than the information loss between input $X_{adv}$ and $Y$. Thus, *a small information loss gets amplified in hidden layers*.

We remark that this theoretical finding is in line with Hong et al. (2020), which states that the difference between adversarial and benign samples is less significant in early layers compared to deeper ones, thus enabling the early exits.

To better prove this point, We investigate the robustness for all the latent spaces in different depths of DNNs by showcasing a small model VGG16 and a deeper model ResNet152. The same perturbation magnitude in the $l_\infty$ norm is enforced for all feature maps before the max pooling layers.

As shown in Table C-1 and Table C-2, latent representations are always more robust than the input irrespective of the depth and dimensionality. Moreover, ASR decreases with the depth of feature maps for both VGG16 and ResNet152.

Table C-1: adversarial robustness of VGG16 as a function of depth with perturbation budget $\epsilon = 0.003$.

| | input (dim×1) | feature 0 (dim×5.33) | feature 1 (dim×2.66) | feature 2 (dim×1.33) | feature 3 (dim×0.66) | feature 4 (dim×0.33) |
|---|---|---|---|---|---|---|
| FGSM | 94.5% | 41.2% | 28.7% | 20.5% | 14.0% | 6.4% |
| PGD | 98.8% | 50.3% | 33.2% | 22.3% | 14.5% | 6.5% |

Table C-2: adversarial robustness of ResNet152 as a function of depth with perturbation budget $\epsilon = 0.003$.

| | input (dim×1) | feature 0 (dim×5.33) | feature 1 (dim×5.33) | feature 2 (dim×2.66) | feature 3 (dim×1.33) | feature 4 (dim×0.66) |
|---|---|---|---|---|---|---|
| FGSM | 71.9% | 53.6% | 53.1% | 35.8% | 34.9% | 1.6% |
| PGD | 88.1% | 65.5% | 66.3% | 43.1% | 43.2% | 1.6% |

## D  SUPPLEMENTARY RESULTS FOR ADVERSARIAL TRAINING

Table D-1: Input and Latent robustness of ResNet50, FastAT (Wong et al., 2020) and DAT (Zhang et al., 2022a) with perturbation budget $\epsilon = 0.003$.

| Models | PGD ($l_\infty$) input | PGD ($l_\infty$) latent | N-Attack input | N-Attack latent | Triangle Attack input | Triangle Attack latent |
|---|---|---|---|---|---|---|
| ResNet50 | 92.9% | **62.9%** | 61.1% | **45.5%** | 85.8% | **13.0%** |
| FastAT | 7.9% | **3.1%** | 43.3% | **1.0%** | 29.6% | **1.1%** |
| DAT | 14.6% | **1.0%** | 54.5% | **0.0%** | 53.0% | **1.0%** |

To have more comprehensive experiments, we evaluate the input and latent robustness of ResNet50 trained using two recent adversarial training approaches, namely FastAT (Wong et al., 2020) and DAT (Zhang et al., 2022a).

As shown in Table D-1, compared to the original ResNet50, both input and latent robustness are increased after adversarial training. From a theoretical perspective, this is because adversarial training introduce more samples $n$, hence reducing the error bound $\mathcal{O}(|\mathcal{T}||\mathcal{Y}|/\sqrt{n})$ in Equation 3.

The ASR of PGD in input space decreases significantly for both FastAT and DAT while the ASR of the other two attacks decreases marginally. This is because such adversarial training approaches use PGD for data augmentation, while distribution shifts from other attacks are missing.

On the other hand, attacks in the input space constantly result in a larger ASR than those in the latent space. As the Equation 10 does not depend on the sample size $n$, our second theoretical finding that latent representations are more robust than the input still holds for the adversarially-trained DNNs.

