# OpenReview forum: "Adversarial Machine Learning in Latent Representations of Neural Networks"
_ICLR.cc/2024/Conference — Submitted to ICLR 2024_

### Official Review · Reviewer_4MyR · 2023-10-27

**Soundness:** 2 fair
**Presentation:** 2 fair
**Contribution:** 2 fair
**Rating:** 3
**Confidence:** 4

**Summary:**

This paper uses information bottleneck (IB) analysis to study adversarial robustness in the split learning setting, consisting of a data encoder that gives the latent representations, followed by a local deep neural network (DNN) that takes the latent presentation for subsequent inference. The results show that the compressed latent representations can reduce the success rate of adversarial attacks, as also indicated by the theory.

**Strengths:**

1. Use IB to study adversarial robustness is a good angle.
2. The paper is well-written and easy to read

**Weaknesses:**

I have several major concerns about the technical novelty and empirical evaluation.

1. On the claim that "assuming the same level of information distortion, latent features are always more robust than input representations", does this hold even if the latent embedding dimension is larger than the input dimension?  If yes, why it can be more robust? Moreover, if the latent embedding dimension is lower than the input dimension, then it has been proven in the ICML 2019 paper "First-order Adversarial Vulnerability of Neural Networks and Input Dimension" that the minimal adversarial perturbation scales inversely with the data dimension. So if we treat the latent representations as the new "data" and they have lower dimensions than the raw inputs, the new insights are not clear.

2. The assumption on the same level of information distortion seems very strong and lacks justification. It's also not clear what is the threat model (what the attacker can do) that leads to a latent representation $T_{adv}$.

3. The evaluated attacks are naive input perturbation attacks, and no adaptive attacks that take into account modifying the latent representations were studied. It should be easy to add a regularizer to attack objectives to encourage finding adversarial examples that share very similar (or very different) latent representations as the original data, and therefore the claim on improved robustness may not hold.

**Questions:**

1. W.r.t. to W1, is there any implicit assumption that the latent embedding dimension is smaller than the input dimension? If so, the improved robustness is a direct consequence of the ICML 2019 result.

2. How to justify the assumption of "the same level of information distortion"? Does $T_{adv}$ hold for any arbitrary threat model?

3. Does the result of improved robustness still hold against adaptive attacks, where the attacker can have access to the latent representations?

---

> ### Author Response · Authors · 2023-11-18
>
> ## **Regarding Weakness 1 & Question 1**
>
> > On the claim that "assuming the same level of information distortion, latent features are always more robust than input representations", does this hold even if the latent embedding dimension is larger than the input dimension?
>
> We thank the reviewer for this comment. To this end, we remark that Theoretical Finding #2 indeed holds when the latent embedding dimension is larger than the input dimension. Our experiments shown in Figure 4 include the original VGG and ResNet, where the space is respectively $\times 5.33$, $\times 2.66$ larger.
>
> To better prove our point, we are actively working on a new experiment with respect to the depth of latent space in the revision. We present some preliminary results in Table 1 and Table 2.
>
> ---
>
> **Table 1: Adversarial robustness of ResNet152 as a function of depth with perturbation budget $\epsilon = 0.003$**
>
> | Attack | Input | FeatureMap0 | FeatureMap1 | FeatureMap2 | FeatureMap3 | FeatureMap4 |
> | :-------: | :-------: | :------: | :-------: | :-------: | :-------: | :--------: |
> | dimension | x1 | x5.33 | x5.33 | x2.66 | x1.33 | x0.66 |
> | FGSM | 71.9% | 53.6% | 53.1% | 35.8% | 34.9% | 1.6% |
> | PGD | 88.1% | 65.5% | 66.3% | 43.1% | 43.2% | 1.6% |
>
> ---
>
> **Table 2: Adversarial robustness of VGG16 as a function of depth with perturbation budget $\epsilon = 0.003$**
>
> | | Input | FeatureMap0 | FeatureMap1 | FeatureMap2 | FeatureMap3 | FeatureMap4 |
> | :-------: | :-------: | :------: | :-------: | :-------: | :-------: | :-------: |
> | dimension | x1 | x5.33 | x2.66 | x1.33 | x0.66 | x0.33 |
> | FGSM | 94.5% | 41.2% | 28.7% | 20.5% | 14.0% | 6.4% |
> | PGD | 98.8% | 50.3% | 33.2% | 22.3% | 14.5% | 6.5% |
>
> ---
>
> The experiment is based on a smaller model VGG16 and a deeper model ResNet152. Same level of $l_\infty$ norm distortions are introduced to all feature maps before max pooling layers. As shown in Table 1 and Table 2, all the latent feature maps are more robust than the input regardless of dimensionality and depth. Furthermore, ASR reduces with respect to the depth of latent space, which is in line with our Theoretical Finding #2.
>
>
> >  If yes, why it can be more robust?
>
> This finding holds because it is derived directly from the chain rule of mutual information which does not depend on cardinality $|\mathcal{T}|$.
>
> > it has been proven in the ICML 2019 paper "First-order Adversarial Vulnerability of Neural Networks and Input Dimension" that the minimal adversarial perturbation scales inversely with the data dimension.
>
> We thank the reviewer for bringing [1] to our attention. We want to address the key difference between the work [1] and our Theoretical Finding #1. In our Theoretical Finding #1, the robustness is jointly determined by its upper bound $I^*(Y;T)$ and $\mathcal{O}( |\mathcal{T}||\mathcal{Y}| / \sqrt{n})$. Although $\mathcal{O}( |\mathcal{T}||\mathcal{Y}| / \sqrt{n})$ indicates that a larger cardinality will result in less robustness, which is in line with [1], $T$ is a DNN-specific space depending on the DNN  architecture design. In stark opposition, Theoretical Finding #1 addresses the variance-bias tradeoff in the distributed DNN design, which is also supported by our experiments as shown in Figure 6. Moreover, it also explains why latent space is more robust even if it has larger cardinality, since the latent robustness depends both on the cardinality and $I^*(Y;T)$.
>
> ## **Regarding Weakness 2 & Question 2**
>
> > The assumption on the same level of information distortion seems very strong and lacks justification
>
> We thank the reviewer for this insightful comment. We remark that such constraints on information distortion are necessary to formally analyze robustness of distributed DNNs and provide a fair comparison with attacks in the input space. Moreover, although there is much work focused on unconstrained attacks –  such as [2, 3, 4] —,  the constrained distortions are still a popular way to evaluate robustness [5, 6]. Since this is the first work to investigate this topic, we aim at formally analyzing the robustness with information constraints. However, we also point out that in real-world scenarios the attacks in the latent space can have a different level of information than the input. We will clarify this crucial aspect in the final version of the paper.
>
> > It's also not clear what is the threat model (what the attacker can do) that leads to a latent representation $T_{adv}$
>
> In a distributed DNN scenario, latent representations are sent to the local DNN that can be exposed to attackers. We refer the reviewer to Figure 1 for a visual depiction of the attack. We assume that attackers can directly add perturbations to latent representations in the same manner as attacks performed in the input space. Thus, $T_{adv}$ denotes the adversarial latent representation generated by attackers. For a detailed attack model formulation, we kindly refer the reviewer to Section 4.1.

---

> > ### Author Response · Authors · 2023-11-18
> >
> > ## **Regarding Weakness 3 & Question 3**
> >
> > > The evaluated attacks are naive input perturbation attacks, and no adaptive attacks that take into account modifying the latent representations were studied.
> >
> > We sincerely thank the reviewer for pointing out adaptive attacks. For a fair comparison, in this paper we consider attackers in latent space as acting in the same manner as those attacking in input space. For this reason, we applied the attacks already present  in literature to the latent representations. Any attack that will lead to different processes between input and latent space are considered out of the scope of our paper since they deserve a separate and dedicated investigation.
> >
> > We remark that adaptive attacks are usually proposed to correctly evaluate novel defense strategies that can tackle attackers by shattered gradient or randomness [7, 8]. Since our focus is to analyze the robust performance of distributed DNNs, instead of providing defenses against known attacks, we believe this new direction deserves a separate investigation.
> >
> > We also point out that much better attacks and defenses specifically tailored for distributed DNNs can be formulated by achieving a fundamental understanding of robustness in distributed DNNs. Given that we are the first to investigate the robustness in distributed DNNs, we would like to limit our scope to well established attack algorithms for understanding the distributed DNNs. We are actively working on novel attack and defense as part of ongoing efforts.
> >
> > ---------------------------------
> >
> > ### **Reference**:
> >
> > [1] Simon-Gabriel, Carl-Johann, et al. "First-order adversarial vulnerability of neural networks and input dimension." International conference on machine learning. PMLR, 2019.
> >
> > [2] Bhattad, Anand, et al. "Unrestricted adversarial examples via semantic manipulation." arXiv preprint arXiv:1904.06347 (2019).
> >
> > [3] Hosseini, Hossein, and Radha Poovendran. "Semantic adversarial examples." Proceedings of the IEEE Conference on Computer Vision and Pattern Recognition Workshops. 2018.
> >
> > [4] Duan, Ranjie, et al. "Adversarial laser beam: Effective physical-world attack to dnns in a blink." Proceedings of the IEEE/CVF Conference on Computer Vision and Pattern Recognition. 2021.
> >
> > [5] Dong, Yinpeng, et al. "Benchmarking adversarial robustness on image classification." proceedings of the IEEE/CVF conference on computer vision and pattern recognition. 2020.
> >
> > [6] Croce, Francesco, et al. "Robustbench: a standardized adversarial robustness benchmark." arXiv preprint arXiv:2010.09670 (2020).
> >
> > [7] Athalye, Anish, Nicholas Carlini, and David Wagner. "Obfuscated gradients give a false sense of security: Circumventing defenses to adversarial examples." International conference on machine learning. PMLR, 2018.
> >
> > [8] Tramer, Florian, et al. "On adaptive attacks to adversarial example defenses." Advances in neural information processing systems 33 (2020): 1633-1645.

---

> ### Comment · Reviewer_4MyR · 2023-11-21
>
> I thank the reviewer for providing the responses. Although some of my concerns have been partially addressed, some major concerns remain:
> 1. Lack of robustness evaluation on adaptive attacks - without this experiment, it's hard to argue, even numerically, that the analysis and the assumptions hold in practice. As I pointed out, many earlier defenses based on latent representation regularization were later broken by adaptive attacks. This setting should apply to the considered distributed DNN setting as well.
> 2. I appreciate the new experiments from authors. However, it is implemented with a very small $\epsilon$ budget and on simple attacks like FGSM and PGD. It would be more convincing to try on larger $\epsilon$ values with AutoAttack.

---

> > ### Author Response · Authors · 2023-11-21
> >
> > > many earlier defenses based on latent representation regularization were later broken by adaptive attacks. This setting should apply to the considered distributed DNN setting as well.
> >
> >
> > We are thankful for the reviewer's valuable feedback.
> >
> > We would like to clarify that defenses based on latent regularization are conceptually different from our work. Those defenses usually consider relaxation or randomization in latent space and formulate an optimization problem. In this case, latent information can be used as a lagrangian regularization in the dual problem. Such regularization is easy to cause gradient obfuscation and thus adaptive attacks are needed to correctly evaluate the robustness.
> >
> > However, distributed DNN is not considered a defense strategy against adversarial attacks. A distributed DNN divides itself into different parts for saving computation power on mobile devices. The tail of the DNN deployed on a local server is a model without considering advanced defense approaches and does not contain any non-differentiable blocks or randomness which can mislead gradient attacks. To this end, strong black box attacks such as Square Attack, which is also a part of autoattack, can be good candidates to evaluate the robustness.
> >
> > We would like to clarify that the goal of our work is not to propose novel defense strategies in latent space against adversarial attacks. Rather, our focus is to compare the robustness of the latent space with that of the input against non-adaptive attacks.
> >
> > > However, it is implemented with a very small budget and on simple attacks like FGSM and PGD. It would be more convincing to try on larger values with AutoAttack.
> >
> > We completely agree with the reviewer that the new experiment needs more comprehensive evaluation with different attacks and multiple perturbation budgets. We are currently working on that. Note that we are working on ImageNet and several strong black-box attacks which require large computational time for finding the effective adversarial samples. We first updated the results of FGSM and PGD because gradient attacks are the most rapid algorithm for us to get a proof of concept during rebuttal. We thank the reviewer for pointing out autoattack, and we are going to include the results in the near future.

---

> > > ### Author Response · Authors · 2023-11-21
> > >
> > > In addition, although adaptive attacks cannot be applied to our settings, we considered a wide range of black-box and white-box attacks to develop a comprehensive understanding of robustness in latent representations. For example, we have multiple gradient-based attacks, boundary-based attacks such as sign-opt, triangle attacks and also attacks such as N-attack, Square and Evolutionary attacks that are based on heuristic search. We would like to point out that these are similar to the work autoattack, which proposed using multiple attacks to better evaluate robustness. To be precise, in autoattack, 2 gradient-based APGD attacks, one heuristic search Square attack, and one boundary-based FAB attack are combined to generate effective adversarial examples. The difference here is that, instead of reporting the worst-case accuracy, we are reporting the attack success rate for each attack separately.

---

> > > > ### Comment · Reviewer_4MyR · 2023-11-22
> > > >
> > > > I thank the authors again for the clarification. I agree that the proposed setting is not a defense, but a claim that the considered distributed DNN setup will enhance robustness.
> > > >
> > > > However, the reason I brought this up is that in this setting, knowing partial information about the embedding is similar to the construction of adaptive attacks. One cannot guarantee that having less information about the entire network will automatically yield improved robustness, especially since the theory relies on the strong assumption of information bottleneck. Adaptive attacks are by far the easiest way I can think of to support the theoretical claim. For example, one can generate adversarial examples that make the local embeddings as close as possible to the original samples, etc.
> > > >
> > > > On attack evaluation, following the norm of adaptive attacks, I also suggest the authors show a complete robustness accuracy versus attack strength ($\epsilon$) plot. This is also strong empirical evidence for inspecting the robustness claims. See https://arxiv.org/abs/1902.06705 for more details.

---

> > > > > ### Author Response · Authors · 2023-11-22
> > > > >
> > > > > We sincerely thank the reviewer for the suggestion. We believe this can enhance our work, and we are adding new experiments as suggested. However, due to time limit, we cannot provide results immediately. We are going to update the paper in the near future.

---

### Official Review · Reviewer_nH51 · 2023-10-29

**Soundness:** 3 good
**Presentation:** 3 good
**Contribution:** 3 good
**Rating:** 8
**Confidence:** 5

**Summary:**

The paper studies the adversarial robustness of deep neural networks (DNNs) in latent space and compares it with the more common adversarial attacks on DNN inputs. The target models are distributed DNNs where the network is splitted in two parts, one is running on a mobile device and sends the output features to other part of the network which is running in cloud. The authors build on top of Information Bottleneck (IB) theory and show that 1) latent features of DNNs are more robust to adversarial attacks than inputs and 2) the smaller the latent dimension, the more difficult it is for an adversary to attack the network successfully. Their results on a wide variety of attacks and networks support the theoretical hypothesis.

**Strengths:**

The paper is easy to follow, has good background sections and exploits the work of Shamir et al. 2010 (equation 3) in a natural way. The experiments are performed on a wide range of adversarial attacks, such as gradient-based, score-based and decision based, as well as white-box and black-box attacks. The results are clear and in line with the theoretical hypothesis.

**Weaknesses:**

There are a few weaknesses that I would like to point out:
1. the paper does not take into consideration attacking the latent representations of a DNN that was adversarially trained. To the best of my knowledge, the adversarial training [1] has an impact over the latent representations of a neural network and under certain settings (a perturbation budget $\epsilon$ not too large) the network is robust to adversarial input perturbations. I would be interested in the robustness of adversarially trained networks, would it be possible to perform such an experiment? I believe the paper is not complete without this experiment and I would really appreciate if you could include this.

References:

[1] **TOWARDS DEEP LEARNING MODELS RESISTANT TO ADVERSARIAL ATTACKS**, available at **https://openreview.net/forum?id=rJzIBfZAb**

**Questions:**

1. Related to the weakness point: how do you think the adversarial training would change the success rate of these attacks?
2. recently there was another adversarial attack introduced for slightly more particular DNNs architectures with multi-exits (or early-exits), called DeepSloth [2], which aims to make the early-exits ineffective. They show that this attack changes the latent representation of DNNs (for example, Figure 3 in the paper) to actually create delay.
3. Did think about analyzing the latent features robustness in LLMs? What do you think the challenges would be?
4. In Section 6 you mention about defense mechanisms for adversarial attack on latent features. What would be the key element in designing defenses for this attack?

References:

[2] **A Panda? No, It's a Sloth: Slowdown Attacks on Adaptive Multi-Exit Neural Network Inference**, available at **https://openreview.net/forum?id=9xC2tWEwBD**

---

> ### Author Response · Authors · 2023-11-18
>
> ## **Regarding Weakness 1 & Question 1**
>
> > The paper does not take into consideration attacking the latent representations of a DNN that was adversarially trained.
>
> We thank the reviewer for pointing this out. We have developed a new experiment to evaluate the input and latent robustness of two adversarial training approaches presented in recent work [1] and [2] in Table 1.
>
> ---
>
> **Table 1: Robustness of ResNet50, DAT [1] and FastAT [2] with perturbation budget $\epsilon = 0.003$**
>
> | | PGD | | N-Attack | | Triangle | |
> | :------- | :-------: | :------: | :-------: | :-------: | :-------: | :--------: |
> | Model | Input | Latent |  Input | Latent | Input | Latent |
> |ResNet50 | 92.9% | 62.9% | 61.1% | 45.5% | 85.8% | 13.0% |
> | DAT | 14.6% | 1.0% | 54.5% | 0.0% | 53.0% | 1.0% |
> | FastAT | 7.9% | 3.1% | 43.3% | 1.0% | 29.6% | 1.1% |
>
> ---
>
> In Table 1, DAT and FastAT are pretrained ResNet50 with different adversarial training approaches. We evaluate the robustness in both input and latent space as we did with the original ResNet50. The results show that latent representations of adversarially trained DNNs are also more robust than input, which is in line with our theoretical findings.
>
>
> > how do you think the adversarial training would change the success rate of these attacks?
>
> We have evaluated this aspect from an experimental standpoint in the table above. From a theoretical perspective, while the adversarial training should be able to improve the robustness in both input and latent space to some extent, it will not affect our Theoretical Finding #2. Indeed, in our Theoretical Finding #1, the upper bound is defined by a term $\mathcal{O}( |\mathcal{X}||\mathcal{Y}| / \sqrt{n})$, where n is the number of samples. Adversarial training actually introduces more samples, resulting in a smaller $\mathcal{O}( |\mathcal{X}||\mathcal{Y}| / \sqrt{n})$ as well as a larger upper bound in robustness. We also point out that our Theoretical Finding #2 holds irrespective of the sample size n, therefore adversarial training will not impact this finding. We are going to include these explanations in the final version of the manuscript.
>
> ## **Regarding Question 2**
>
> > Recently there was another adversarial attack introduced for slightly more particular DNNs architectures with multi-exits (or early-exits)
>
> We thank the reviewer for bringing [3] to our attention. We point out that [3] can be explained with information bottleneck theory which is related to our Theoretical Finding #2. We explain it as follow:
>
> It is pointed out in [3] that conventional adversarial attacks cannot slow down the early-exit networks.  Since we proposed to measure distortion with residual information $I(X;Y|T)$, the residual information of adversarial samples $I(X_{adv};Y|T’)$ will be larger than the residual information of benign samples $I(X;Y|T)$, i.e.,
>
> $I(X_{adv};Y|T') \geq I(X;Y|T)$
>
> From Equation 8 in our paper,  it follows that
>
> $I(X;Y) - I(X_{adv};Y) \leq I(T;Y) - I(T’;Y).$
>
> In other words, the information loss between latent $T’$ and ground-truth $Y$ is larger than the information loss between input $X_{adv}$ and $Y$. Thus, a small information loss gets amplified in hidden layers, which is consistent with Figure 3 in [3].
>
> It follows that in early-exit models, the adversarial samples and benign samples will have less distortion in early layers, and thus early exits can be activated.
>
> We thank the reviewer for pointing out [3] which could help us enhance our theoretical aspect. We are working on a revision to include [3] and the discussion.

---

> ### Author Response · Authors · 2023-11-18
>
> ## **Regarding Question 3**
>
> > Did you think about analyzing the latent features robustness in LLMs?
>
> We believe the adversarial robustness of latent features in LLMs is an intriguing and extremely timely topic, and we are planning to have a separate investigation into this matter.
>
> > What do you think the challenges would be?
>
> We believe one big challenge could be the intrinsic difference between textual and image data. For example, textual data are non-differentiable due to their semantic nature, while image data are differentiable RGB values. As a result, the problem formulation cannot be directly applied to LLMs.
>
>
> ## **Regarding Question 4**
>
> > What would be the key element in designing defenses for this attack?
>
> We believe the latent representations intrinsically have more semantic information compared to input. And there is evidence that adversarial input and benign input have different distributions in hidden layers [4]. We believe this can also be generalized to latent representations. In order to develop defense specifically for latent representations, we believe one key element is to understand the semantic meaning of the distribution change caused by adversarial perturbations.
>
> ---
>
> ### **Reference**:
>
> [1] Zhang, Gaoyuan, et al. "Distributed adversarial training to robustify deep neural networks at scale." Uncertainty in Artificial Intelligence. PMLR, 2022.
>
> [2] Wong, Eric, Leslie Rice, and J. Zico Kolter. "Fast is better than free: Revisiting adversarial training." arXiv preprint arXiv:2001.03994 (2020).
>
> [3] Hong, Sanghyun, et al. "A panda? no, it's a sloth: Slowdown attacks on adaptive multi-exit neural network inference." arXiv preprint arXiv:2010.02432 (2020).
>
> [4] Galloway, Angus, et al. "Batch normalization is a cause of adversarial vulnerability." arXiv preprint arXiv:1905.02161 (2019).

---

> ### Comment · Reviewer_nH51 · 2023-11-20
>
> I would like to thank authors for addressing my comments.
>
> **About Table 1.** I think these results should be added to the paper because I believe they will add value to the paper for the simple fact that attacks on latent features are even less successful than attacks on inputs for adversarially trained networks.
>
> I agree with the authors' explanations to my questions, they are intuitive and I really appreciate the effort you put into this. As a result, I will increase my score to 8.

---

### Official Review · Reviewer_z2Hg · 2023-10-31

**Soundness:** 3 good
**Presentation:** 3 good
**Contribution:** 2 fair
**Rating:** 6
**Confidence:** 3

**Summary:**

A distributed DNN can be regarded as a combination of two parts, namely, a mobile DNN and a local DNN, respectively. The mobile DNN is trained to learn the latent representations, which can reduce the amount of data that will be transmitted but suffers the risk of being attacked. Along this side, this paper investigates the robustness of the latent representations. Based on information theory, this paper claims that: 1) latent features are always more robust than input representations, and 2) the feature dimensions and the generalization capability of the DNN determine the adversarial robustness. Extensive experiments on ImageNet-1K are conducted to support the claims, considering 6 different DNN architectures, 6 different ways for distributed DNN, and 10 different adversarial attacks.

**Strengths:**

1. This paper is well-written, with clear explanations and illustrations. Section 2 is comprehensive, and someone interested in those related topics can learn from the article.
2. Based on Information theory, Section 3 provides a thorough theoretical analysis. Besides, the theoretical conclusions are supported by the experimental results in Section 4 with detailed experimental settings.

**Weaknesses:**

In the Conclusion section, this article claims that ``This paper has investigated adversarial attacks to latent representations of DNNs for the first time``. Through the lens of distributed DNNs, this work may be the first one, as it claims. However, I am concerned about whether the distributed DNNs are a necessary background as the motivation to investigate the problem. Since I think the local DNN and mobile DNN are very like the architecture of an autoencoder, and there are many works about the adversarial robustness of autoencoders.

**Questions:**

As mentioned in the Weaknesses, my questions/concerns are mainly about the differences between the distributed DNNs and the autoencoders.
1. If we compare the architecture between a distributed DNN and an autoencoder, I think the local DNN is very much like the encoder part, and the mobile DNN is very much like the decoder part. Can I compare them like this?
2. If yes, I think some works have studied the adversarial robustness of the latent features, e.g., [1].
3. Therefore, I am a bit curious about whether distributed DNNs are a necessary background as the motivation to investigate the adversarial robustness of the latent features.

---
[1] Espinoza-Cuadros, F. M., Perero-Codosero, J. M., Antón-Martín, J., & Hernández-Gómez, L. A. (2020). Speaker de-identification system using autoencoders and adversarial training. arXiv preprint arXiv:2011.04696.

---

> ### Author Response · Authors · 2023-11-18
>
> ## **Regarding Weakness & Questions**
> > However, I am concerned about whether the distributed DNNs are a necessary background as the motivation to investigate the problem. Since I think the local DNN and mobile DNN are very like the architecture of an autoencoder, and there are many works about the adversarial robustness of autoencoders.
>
> We agree with the reviewer that distributed DNNs have an encoder-decoder structure similar to autoencoders, and we appreciate the reviewer pointing out adversarial autoencoders. However, there is a stark distinction between robust distributed DNNs and robust autoencoders. In robust autoencoder, adversarial training is used to enforce the encoder to learn adversarially robust latent representations – for example, for speaker de-identification [1] or outlier detection [2]. Such latent representations are not considered to be exposed to attackers thus the “robustness” in the robust autoencoder context is actually “semantically meaningful representations”.
>
> In our case, we consider attacks adding perturbations directly to the latent space. The scenario considered is extremely relevant to mobile scenarios since distributed DNNs have been proven to be extremely effective in reducing network load while preserving accuracy [3, 4, 5]. In this case, latent representations are exposed to the attacker, which can lead to performance degradation. To this end, the “robustness” in latent representation means “adversarially robust to attacks”, which is a remarkably different concept than robust autoencoders.
>
> To the best of the authors’ knowledge, there is no such prior work to investigate adversarial perturbation in latent representation. Thus, we would like to highlight the significance and timeliness of our work in the context of distributed DNN.
>
> We appreciate the constructive feedback to help us clarify our novelty and contribution. We are actively working on a revision that includes the suggested work [1] and clarification.
>
> ---------------------------------
>
> Reference:
>
> [1] Espinoza-Cuadros, Fernando M., et al. "Speaker de-identification system using autoencoders and adversarial training." arXiv preprint arXiv:2011.04696 (2020).
>
> [2] Salehi, Mohammadreza, et al. "Arae: Adversarially robust training of autoencoders improves novelty detection." Neural Networks 144 (2021): 726-736.
>
> [3] Matsubara, Yoshitomo, et al. "Bottlefit: Learning compressed representations in deep neural networks for effective and efficient split computing." 2022 IEEE 23rd International Symposium on a World of Wireless, Mobile and Multimedia Networks (WoWMoM). IEEE, 2022.
>
> [4] Matsubara, Yoshitomo, et al. "Supervised compression for resource-constrained edge computing systems." Proceedings of the IEEE/CVF Winter Conference on Applications of Computer Vision. 2022.
>
> [5] Shao, Jiawei, and Jun Zhang. "Bottlenet++: An end-to-end approach for feature compression in device-edge co-inference systems." 2020 IEEE International Conference on Communications Workshops (ICC Workshops). IEEE, 2020.

---

> > ### Comment · Reviewer_z2Hg · 2023-11-22
> > **Thanks to the Authors for the Further Explanation**
> >
> > Thanks to the authors clarifying the differences between the distributed DNNs and autoencoders. I would like to keep my original score of 6.

---

### Official Review · Reviewer_an5M · 2023-11-01

**Soundness:** 3 good
**Presentation:** 3 good
**Contribution:** 2 fair
**Rating:** 6
**Confidence:** 3

**Summary:**

This paper studies the robustness of distributed DNN against adversarial attack theoretically and experimentally. The authors analyze the robustness of latent features using information bottleneck theory and prove that latent space perturbations are always less effective than input space perturbations. Empirically, the authors conduct extensive experiments to verify their theoretic findings from multiple perspectives.

**Strengths:**

The paper is well written in general. The theoretical analysis from information bottleneck perspective is interesting and solid. Most of the claims are supported by ample experimental analysis.

**Weaknesses:**

1. In this paper, the attacker only has access to the latent representation provided by the mobile DNN from a mobile phone. While the authors successfully demonstrate that attacks on these latent representations exhibit a lower Attack Success Rate (ASR) than those on raw images—given the same level of information distortion—the appropriateness of imposing an identical distortion level in this context needs more clarification. In discussions around the adversarial robustness of image input, constraints are placed on distortion levels to ensure modifications are imperceptible to the human eye, yet potent enough to deceive the classifier [1]. However, when considering distortions applied to latent features, the rationale for enforcing the same constraint level is less clear. Given that the transmission in this scenario occurs between a mobile device and a cloud computer, with no human observer in the loop, the attacker might as well nullify all latent features, potentially achieving an ASR close to 100%. Comparing the ASR between attacks on raw input images (which are observable by humans) and attacks on latent features (processed by a cloud computer) under an equivalent distortion level seems illogical.

2. Although the authors posit that this study on the robustness of distributed DNNs in the face of adversarial actions is novel, I find the concept markedly similar to existing works on attacks targeting intermediate layers or latent features [2]. I would appreciate it if the authors could highlight the distinctions between their work and prior research, and attempt to apply the attack methods delineated in [2] where feasible.

3. I am confused about two terminologies in the paper: feature compression and bottleneck. Their relationship is not clear to me. In Table 1, it seems that only the first two feature compression methods contain a bottleneck layer. However, in Section 4.2 - DNN architectures, the authors write, '...the feature compression layer (i.e., the 'bottleneck').' Additionally, the authors use the same bottleneck design as Matsubara et al. (2022b) and denote the new architectures with feature compression as Resnet50-fc, etc. However, the specific design mentioned in Matsubara et al. (2022b) does not seem to belong to any of the six feature compression approaches in Table 1. Table A-3 is also confusing, as I cannot understand what the authors mean by 'JC and QT are DNNs without a bottleneck,' while JC and QT are feature compression approaches, and the authors claim that feature compression is the same as bottleneck.

4. The experimental results in Section 5.3 need further explanation. ResNet152-fc with 12 channels achieves a validation accuracy of 77.47%, while ResNet152-fc with 3 channels achieves 70.01% accuracy. On the middle of page 9, the statement 'decreases to 7.47%' should be corrected to 'decreases by 7.46%.' However, the fact that I∗(Y ; T) decreases by 7.46% cannot explain why the ASR increases by a much larger percentage than 7.46% in Table A-4. For example, when \epsilon=0.003, the ASR of PGD_2 increases by 19.9% when transitioning from 12 channels to 3 channels. According to the inequality in Key Theoretical Finding #1, since O(|T||Y|/√n) is smaller when transitioning from 12 channels to 3 channels, the ASR difference should be less than the difference in I∗(Y ; T), which is 7.46%. Please provide a detailed explanation.

[1] Kurakin, Alexey, Ian J. Goodfellow, and Samy Bengio. "Adversarial examples in the physical world." Artificial intelligence safety and security. Chapman and Hall/CRC, 2018. 99-112.
[2] Yu, Yunrui, Xitong Gao, and Cheng-Zhong Xu. "Lafeat: Piercing through adversarial defenses with latent features." CVPR. 2019.

**Questions:**

My primary concern is related to the validity of the problem setting presented in this paper (See weakness 1). Although the theoretical findings are intriguing and the experimental data is comprehensive, there is still uncertainty regarding the significance of defining the robustness of distributed Deep Neural Networks (DNNs) in the proposed manner.

---

> ### Author Response · Authors · 2023-11-18
>
> ## **Regarding Weakness 1**
>
> > the appropriateness of imposing an identical distortion level in this context needs more clarification
>
> We thank the reviewer for this insightful comment. We point out that estimating the distortion is necessary to formally compare the robustness of the DNN against adversarial attacks in the latent space with respect to attacks in the input space. Specifically, to compare input space and latent space attacks, we assume an identical adversarial effort in both input and latent space through the $l_p$ norm constraint. We point out that the $l_p$ norm is routinely used as a metric to characterize such effort, as it is used in existing work, for example [1, 2, 3]. Moreover, we also point out that in real-world scenarios the attacks in the latent space can (and most likely, will) have a different level of information than attacks in the input space. We will clarify this crucial aspect in the final version of the paper.
>
> > with no human observer in the loop, the attacker might as well nullify all latent features, potentially achieving an ASR close to 100%
>
> We strongly agree with the reviewer that without a human observer in the loop, unconstrained perturbations in latent space can lead to ~100% ASR. However, we also believe latent representations intrinsically contain more semantic information than the input, which may result in easier outlier detection. For example, image data usually has redundant information such as background. Therefore, changing one pixel in the background might not be perceptible by humans. Conversely, since semantic representations capture only the sufficient statistics of the input, changing one element of them can result in invalid interpretations (take one-hot encoding as an extreme case example). As such, an outlier detection system which can estimate the perturbation magnitude in latent space can play a similar role as the human observer in input space. We will definitely delve deeper into this intriguing aspect in our future work as it deserves a separate and exhaustive investigation.
>
> ## **Regarding Weakness 2**
>
> > I would appreciate it if the authors could highlight the distinctions between their work and prior research
>
> We appreciate the reviewer bringing [4] to our attention. We point out that paper [4] uses the latent representation as a help to craft the adversarial input. In stark opposition, our work focuses on the perturbation of the latent space itself. We are working on a revision to address the distinction between our work and suggested work [4].
>
> > and attempt to apply the attack methods delineated in [4] where feasible.
>
> We thank the reviewer for suggesting to compare our paper to [4], which we definitely plan to do in future work.
>
> ## **Regarding Weakness 3**
>
> > I am confused about two terminologies in the paper: feature compression and bottleneck.
>
> We are sorry for the confusion. In the literature, we refer to “bottleneck” as a neural network architecture that can be trained in different ways to reduce the dimensionality of feature representations. For this reason, we present different training strategies for bottlenecks – Supervised Compression (SC), Knowledge Distillation (KD), Entropic Student (ES) and BottleFit (BF).
>
> Conversely, other feature compression approaches such as JEPG Compression (JC) and Quantization (QT) compress features by digitizing the feature map in its frequency domain and original domain respectively. These approaches compress the data size but are not able to reduce the dimensionality of the latent space.
>
> In short, feature compression can be done in different ways, compressing the dimensionality (bottleneck) and compressing the data size (JC and QT).
>
> > Additionally, the authors use the same bottleneck design as Matsubara et al. (2022b) and denote the new architectures with feature compression as Resnet50-fc, etc. However, the specific design mentioned in Matsubara et al. (2022b) does not seem to belong to any of the six feature compression approaches in Table 1.
>
> Thank you for your remark. Since there can be different bottleneck architectures, we choose the same architecture that is used in [5] which is a benchmark paper. Notice that similar to our work, [5] did not present the architecture but different training strategies as different approaches. However, in their codebase, they use a consistent bottleneck design.
>
> > Table A-3 is also confusing, as I cannot understand what the authors mean by 'JC and QT are DNNs without a bottleneck,'
>
> We are sorry for the confusion. Since JC and QT are not related to the DNN architecture, such approaches can be add-ons for any DNN with or without bottleneck to reduce the communication overhead. However, as we want to verify the point that dimensionality helps to enhance latent robustness, we choose JC and QT without bottlenecks in our experiment for comparison.

---

> > ### Author Response · Authors · 2023-11-18
> >
> > ## **Regarding Weakness 4**
> >
> > >  On the middle of page 9, the statement 'decreases to 7.47%' should be corrected to 'decreases by 7.46%.'
> >
> > We thank the reviewer for pointing out the mistake and we are actively working on the revision.
> >
> > > Please provide a detailed explanation
> >
> > It is a very good observation that ASR increases by a much larger percentage than the decreased end-to-end performance. This is because the Theoretical Finding #1 describes an upper bound of the robustness which can be considered as the best case scenario. However, adversarial attack algorithms are trying to find the worst case that can fool the DNN. In other words, attackers aim to approach the lower bound of robustness. Thus, the smaller $\mathcal{O}( |\mathcal{T}||\mathcal{Y}| / \sqrt{n})$ is not necessarily equivalent to a less ASR increase.
> >
> > ---------------------------------
> >
> > ### **Reference**:
> >
> > [1] Dong, Yinpeng, et al. "Benchmarking adversarial robustness on image classification." proceedings of the IEEE/CVF conference on computer vision and pattern recognition. 2020.
> >
> > [2] Croce, Francesco, et al. "Robustbench: a standardized adversarial robustness benchmark." arXiv preprint arXiv:2010.09670 (2020).
> >
> > [3] Carlini, Nicholas, et al. "On evaluating adversarial robustness." arXiv preprint arXiv:1902.06705 (2019).
> >
> > [4] Yu, Yunrui, Xitong Gao, and Cheng-Zhong Xu. "Lafeat: Piercing through adversarial defenses with latent features." Proceedings of the IEEE/CVF Conference on Computer Vision and Pattern Recognition. 2021.
> >
> > [5] Matsubara, Yoshitomo, et al. "SC2: Supervised compression for split computing." arXiv preprint arXiv:2203.08875 (2022).

---

> ### Comment · Reviewer_an5M · 2023-11-21
> **Increase my score 6**
>
> I appreciate the detailed explanations and great efforts made by the authors. Thus, I increased my score to 6. However, as mentioned in the reply, the explanations need numerical experiments to support the claims, which can give strong motivation of this work.

---

> > ### Author Response · Authors · 2023-11-21
> > **Response to Reviewer an5M**
> >
> > > However, as mentioned in the reply, the explanations need numerical experiments to support the claims, which can give strong motivation of this work.
> >
> > We would like to thank the reviewer for the valuable feedback. To support our explanations, we have added a new experiment of anomaly detection. We choose 5 classes from the whole ImageNet and get latent representations using the ResNet152-fc model. We have used a latent vector in the training set to train the LOF classifier  [1]. Next,  we have performed anomaly detection on a clean test set, PGD-distorted latent samples and Randomly distorted latent samples, respectively. We have chosen a normal distribution $N(0,1)$ for generating random noise, while the perturbation budget of PGD has been set to 0.01. In this case, the random noise can be considered as large distortions without constraints while PGD can be considered as a constrained perturbation scenario. We report the accuracy output by the DNN classifier and the anomaly ratio output by LOF algorithm in Table 1.
> >
> > As shown in Table 1, the constrained and unconstrained noise can result in 0.4% and 100% anomaly detection rate while PGD also achieved a lower DNN performance. We believe this preliminary result can validate our explanation and we are going to include more comprehensive study in our next revision.
> >
> > -----
> >
> > **Table1: Performance of LOF in latent space with respect to constrained and unconstrained perturbation**
> > | | Accuracy | Anomaly |
> > | :-------: | :-------: | :------: |
> > | Clean | 86.4% | 1.6% |
> > | PGD | 0.4% | 0.4% |
> > | Rand | 9.6% | 100% |
> >
> > ----
> >
> > **Reference**
> >
> > [1] Breunig, Markus M., et al. "LOF: identifying density-based local outliers." Proceedings of the 2000 ACM SIGMOD international conference on Management of data. 2000.

---

> > > ### Comment · Reviewer_an5M · 2023-11-22
> > > **Keep my score**
> > >
> > > Thanks for providing the preliminary experiments. As more comprehensive experiments are needed, I keep my current score.

---

### Official Review · Reviewer_CQZD · 2023-11-05

**Soundness:** 3 good
**Presentation:** 2 fair
**Contribution:** 2 fair
**Rating:** 6
**Confidence:** 4

**Summary:**

This work aims to evaluate the adversarial robustness of distributed DNNs. In this setting, latent representations of the DNNs are communicated among devices, and thus, an adversary could perturb the latent representations instead of the model's input. However, the work claims that attacks on latent representations are less effective than perturbing the input and presents theoretical information bounds supporting this claim. Standard adversarial attacks are then used to test this bound empirically over several distributed DNN architectures, and the results show that the attacks are less successful when employed on latent spaces than on the model's input.

**Strengths:**

The suggested information bound entails that standard adversarial attacks targeting only the latent representation of DNNs would be less effective than those targeting the input. This is relevant not only in distributed DNN but for side-channel attacks as well. Moreover, it greatly aids in evaluating the robustness of distributed DNN, as such attacks arise naturally in this setting.

**Weaknesses:**

1. The experimental setup is lacking and insufficient to support the authors' claims. Only attacks on several distributed DNN architectures were reported. Not only is this insufficient to support the claim that attacks on latent spaces are generally less effective, but it does not explain the phenomenon or the behavior of the suggested information bound.
2. No novel attacks targeting latent spaces were suggested, or even settings in which both the input and latent space are attacked. Without testing such attacks and settings, the robustness of distributed DNN cannot be correctly evaluated.
3. The second key finding of "DNN robustness is intrinsically related to the cardinality of the latent space" is not a phenomenon exclusive to latent spaces. There are several examples of attacks working better on larger input samples such as Imagnet, compared to CIFAR10/100. In addition, the effect of the l_inf norm bound is highly dependent on the input size.

**Questions:**

1. For a correct evaluation of the suggested bound on a given DNN architecture, the experimental settings should present attacks on all the latent spaces in the architecture and not only those available in specific distributed DNN settings. The results should be compared for the depth of the latent spaces in the network and their cardinality. Such experiments consider side-channel attacks on DNN and not only the distributed DNN setting.
2. Attacks targeting explicitly latent spaces should be considered; such attacks should be aware of the specifics of the latent spaces (e.g., depth and cardinality) and make use of them to improve the efficiency of the attack.
3. Adversarial attacks targeting input and latent representations should be considered to evaluate if such a setting presents a greater risk to distributed DNN.
4. As the effectiveness of perturbations depends on the input size, the results should be normalized accordingly.

Post-rebuttal feedback
I am satisfied with authors' clarifications and provided additional evaluation. Hence, my score to 6

---

> ### Author Response · Authors · 2023-11-18
>
> ## **Regarding Weakness 1**
>
> > The experimental setup is lacking and insufficient to support the authors' claims. Only attacks on several distributed DNN architectures were reported.
>
> We appreciate the reviewer’s concern. On the other hand, we point out that it is hardly feasible to investigate each and every distributed DNN architecture since each of them exhibits its own customized design for feature compression and model partitioning.
>
> Moreover, it is well known that neural networks share similar robust and non-robust features in the same dataset [1]. Thus, it has been shown that adversarial samples generated by one model can transfer to another [2, 3, 4].  For this reason, the target of this paper is to provide theoretical proof of the robustness of distributed DNNs and evaluate the soundness of the proof on the 6 most widely-used approaches in recent years derived from the VGG and ResNet family. This was the same procedure followed by other benchmarking papers in the distributed DNN community [5] and adversarial machine learning community [6].
>
> Thus, we think that the combination of theoretical proof and experimental evaluation put forth by the paper provides a strong contribution to the field. We believe that the extensive evaluation of 6 distributed DNN approaches, 6 model architectures as well as 10 attack algorithms is considered a sufficient experimental setup.
>
> ## **Regarding Weakness 2**
>
> > No novel attacks targeting latent spaces were suggested, or even settings in which both the input and latent space are attacked
>
> Thank you for your comment. We point out that proposing new adversarial attacks is out of the scope of the paper, which is instead evaluating the robustness of distributed DNNs to existing attacks, while achieving a fundamental understanding of the robustness of latent spaces.
>
> Moreover, for a fair comparison, we needed to guarantee that each attack we performed could be applied to both input and latent space. Proposing new and advanced attacks specifically tailored to distributed DNNs – for example, the suggested joint input/latent space attacks – is an intriguing direction and deserves a separate and dedicated investigation.
>
> ## **Regarding Weakness 3**
>
> > The second key finding of "DNN robustness is intrinsically related to the cardinality of the latent space" is not a phenomenon exclusive to latent spaces.
>
> We agree that there is other work with similar findings in input space [7]. However, our Theoretical Finding #1 is different from [7] because in our analysis the upper bound is jointly determined by $I^*(Y;T)$ and $\mathcal{O}( |\mathcal{T}||\mathcal{Y}| / \sqrt{n})$. Theoretical Finding #1 focuses on the variance-bias tradeoff in distributed DNN designs, while previous work only focuses on the input dimension. We prove this novel finding with experiments shown in Figure 6.
>
> On the other hand, if we limit our scope to data space $X$, the robustness provided by dataset $X$ can be described by its upper bound $I^*(Y;X)$ and $\mathcal{O}( |\mathcal{X}||\mathcal{Y}| / \sqrt{n})$. Since $X$ represents the dataset, $I^*(Y;X)$ is constant. The adversarial robustness for a given dataset is directly described by the cardinality, which is in line with the findings reported in [7]. For this reason, our analysis from the information perspective can also generalize to input space.
>
> ---------------------------------
> ### **Reference**:
>
> [1] Ilyas, Andrew, et al. "Adversarial examples are not bugs, they are features." Advances in neural information processing systems 32 (2019).
>
> [2] Liu, Yanpei, et al. "Delving into transferable adversarial examples and black-box attacks." arXiv preprint arXiv:1611.02770 (2016).
>
> [3] Dong, Yinpeng, et al. "Evading defenses to transferable adversarial examples by translation-invariant attacks." Proceedings of the IEEE/CVF Conference on Computer Vision and Pattern Recognition. 2019.
>
> [4] Xie, Cihang, et al. "Improving transferability of adversarial examples with input diversity." Proceedings of the IEEE/CVF conference on computer vision and pattern recognition. 2019.
>
> [5] Matsubara, Yoshitomo, et al. "SC2 Benchmark: Supervised Compression for Split Computing." Transactions on Machine Learning Research (2023).
>
> [6] Dong, Yinpeng, et al. "Benchmarking adversarial robustness on image classification." proceedings of the IEEE/CVF conference on computer vision and pattern recognition. 2020.
>
> [7] Simon-Gabriel, Carl-Johann, et al. "First-order adversarial vulnerability of neural networks and input dimension." International conference on machine learning. PMLR, 2019

---

> ### Author Response · Authors · 2023-11-18
>
> ## **Regarding Questions**
>
> > For a correct evaluation of the suggested bound on a given DNN architecture, the experimental settings should present attacks on all the latent spaces in the architecture and not only those available in specific distributed DNN settings.
>
> We strongly agree with the reviewer that to support our general theoretical findings, the attacks should be performed on all the latent spaces in different depths, and we are currently working to include this experiment. We present some preliminary results in Table 1 and Table 2. We are going to add more results in our final revision. As far as the latent cardinality is concerned, we do have an experiment in the current paper which is shown in Figure 6.
>
> ---
>
> **Table 1: Adversarial robustness of ResNet152 as a function of depth with perturbation budget $\epsilon = 0.003$**
>
> |  | Input | FeatureMap0 | FeatureMap1 | FeatureMap2 | FeatureMap3 | FeatureMap4 |
> | :-------: | :-------: | :------: | :-------: | :-------: | :-------: | :--------: |
> | dimension | x1 | x5.33 | x5.33 | x2.66 | x1.33 | x0.66 |
> | FGSM | 71.9% | 53.6% | 53.1% | 35.8% | 34.9% | 1.6% |
> | PGD | 88.1% | 65.5% | 66.3% | 43.1% | 43.2% | 1.6% |
>
> ---
>
> **Table 2: Adversarial robustness of VGG16 as a function of depth with perturbation budget $\epsilon = 0.003$**
>
> | | Input | FeatureMap0 | FeatureMap1 | FeatureMap2 | FeatureMap3 | FeatureMap4 |
> | :-------: | :-------: | :------: | :-------: | :-------: | :-------: | :-------: |
> | dimension | x1 | x5.33 | x2.66 | x1.33 | x0.66 | x0.33 |
> | FGSM | 94.5% | 41.2% | 28.7% | 20.5% | 14.0% | 6.4% |
> | PGD | 98.8% | 50.3% | 33.2% | 22.3% | 14.5% | 6.5% |
>
> ---
>
> We choose one simple model VGG16 and one relatively deeper model ResNet152 for the experiment. In VGG16 and ResNet152, multiple feature extraction blocks which consist of several identical convolutional layers are used to extract features. After each block a max pooling layer is used to reduce the feature dimensions at different depths. We report the ASR in all feature maps generated by feature extraction blocks. As shown in Table 1 and Table 2, irrespective of the depth and dimensionality, latent representations are always more robust than input, which can support our theoretical analysis.
>
> > As the effectiveness of perturbations depends on the input size, the results should be normalized accordingly.
>
> We completely agree that the effectiveness of perturbations depends on the input size. For this reason, in the first version of the paper, we have rescaled the perturbation with respect to cardinality, similar to what is done in Equation 3 in [7]. Please refer to Section 4.1 for more details. Furthermore, we are following the literature where it is not common to normalize the Attack Success Rate.

---

### Author Response · Authors · 2023-11-21
**Global Comment to All Reviewers 1/2**

Dear reviewers,

We would like to sincerely thank you for your valuable feedback, which has helped us improve the quality of our manuscript significantly.

We have uploaded an updated version of our paper, which contains the following major changes:

**1. Clarification of compression approaches in distributed DNNs**

We have added categories in **Table 1** and changed the text accordingly to better summarize compression approaches in the distributed DNN literature. In distributed DNNs, compression can be achieved by both compressing the dimensionality and compressing the data size. While SC, KD, and BF are different training approaches to compress the dimensionality, JC and QT are data compression approaches that do not reduce the dimension. ES is an advanced approach that both compresses the dimension and data size.

**2. Differences with respect to prior work**

In Appendix B, we have updated the distinction with prior work, including adversarial auto-encoders [1-7] and latent-aware adversarial attacks to the input space [8-15]. Specifically, adversarial auto-encoders use adversarial training to get disentangled features that are invariant to unseen data, while latent-aware attacks use latent information as a hint to craft more effective adversarial inputs. Our work separates itself from the above-mentioned literature since we focus on the adversarial perturbations of the latent representation itself. This is because we are considering a distributed scenario in which latent representations can get exposed to attackers during the communication between mobile devices and servers. To this end, we point out the significance and timeliness of our work in the context of distributed DNNs.

**3. Additional explanation of theoretical findings**

In Appendix C, we have clarified that the identical distortion assumption is for formal analysis and fair comparison with adversarial robustness in input space. We have provided a new theoretical explanation that does not require an identical level of distortion constraint. We have addressed the related work [16] and provided new experiments in **Table C-1** and **Table C-2** to support our analysis.

**4. Supplementary experiments**

We have included the evaluation of adversarial robustness for all the latent spaces in different depths in **Table C-1** and **Table C-2**. We have also evaluated the input and latent robustness after performing different adversarial training approaches [17, 18] in **Table D-1**.

---

> ### Author Response · Authors · 2023-11-21
> **Global Comment to All Reviewers 2/2**
>
> **5. Additional Citations**
>
> ---
>
> Adversarial Auto-encoders:
>
> **[1] Espinoza-Cuadros, Fernando M., et al. "Speaker de-identification system using autoencoders and adversarial training." arXiv preprint arXiv:2011.04696 (2020).**
>
> [2] Latif, Siddique, et al. "Multi-task semi-supervised adversarial autoencoding for speech emotion recognition." IEEE Transactions on Affective computing 13.2 (2020): 992-1004.
>
> [3] Sahu, Saurabh, et al. "Adversarial auto-encoders for speech based emotion recognition." arXiv preprint arXiv:1806.02146 (2018).
>
> [4] Salehi, Mohammadreza, et al. "Arae: Adversarially robust training of autoencoders improves novelty detection." Neural Networks 144 (2021): 726-736.
>
> [5] Pidhorskyi, Stanislav, Ranya Almohsen, and Gianfranco Doretto. "Generative probabilistic novelty detection with adversarial autoencoders." Advances in neural information processing systems 31 (2018).
>
> [6] Beggel, Laura, Michael Pfeiffer, and Bernd Bischl. "Robust anomaly detection in images using adversarial autoencoders." Machine Learning and Knowledge Discovery in Databases: European Conference, ECML PKDD 2019, Würzburg, Germany, September 16–20, 2019, Proceedings, Part I. Springer International Publishing, 2020.
>
> [7] Makhzani, Alireza, et al. "Adversarial autoencoders." arXiv preprint arXiv:1511.05644 (2015).
>
> ---
>
> Latent-aware Adversarial Attacks:
>
> **[8] Yu, Yunrui, Xitong Gao, and Cheng-Zhong Xu. "Lafeat: Piercing through adversarial defenses with latent features." Proceedings of the IEEE/CVF Conference on Computer Vision and Pattern Recognition. 2021.**
>
> [9] Sabour, Sara, et al. "Adversarial manipulation of deep representations." arXiv preprint arXiv:1511.05122 (2015).
>
> [10] Inkawhich, Nathan, et al. "Feature space perturbations yield more transferable adversarial examples." Proceedings of the IEEE/CVF Conference on Computer Vision and Pattern Recognition. 2019.
>
> [11] Huang, Qian, et al. "Enhancing adversarial example transferability with an intermediate level attack." Proceedings of the IEEE/CVF international conference on computer vision. 2019.
>
> [12] Inkawhich, Nathan, et al. "Transferable perturbations of deep feature distributions." arXiv preprint arXiv:2004.12519 (2020).
>
> [13] Luo, Cheng, et al. "Frequency-driven imperceptible adversarial attack on semantic similarity." Proceedings of the IEEE/CVF Conference on Computer Vision and Pattern Recognition. 2022.
>
> [14] Mopuri, Konda Reddy, Utsav Garg, and R. Venkatesh Babu. "Fast feature fool: A data independent approach to universal adversarial perturbations." arXiv preprint arXiv:1707.05572 (2017).
>
> [15] Wang, Zhibo, et al. "advpattern: Physical-world attacks on deep person re-identification via adversarially transformable patterns." Proceedings of the IEEE/CVF International Conference on Computer Vision. 2019.
>
> ---
>
> DeepSloth:
>
> **[16] Hong, Sanghyun, et al. "A panda? no, it's a sloth: Slowdown attacks on adaptive multi-exit neural network inference." arXiv preprint arXiv:2010.02432 (2020).**
>
> -------------------
>
> Adversarial Training:
>
> [17] Zhang, Gaoyuan, et al. "Distributed adversarial training to robustify deep neural networks at scale." Uncertainty in Artificial Intelligence. PMLR, 2022.
>
> [18] Wong, Eric, Leslie Rice, and J. Zico Kolter. "Fast is better than free: Revisiting adversarial training." arXiv preprint arXiv:2001.03994 (2020).
>
> ---
>
> Adversarial vulnerability is related to dimensions:
>
> **[19] Simon-Gabriel, Carl-Johann, et al. "First-order adversarial vulnerability of neural networks and input dimension." International conference on machine learning. PMLR, 2019.**

---

### Meta-Review · Area_Chair_9QXf · 2023-12-05

**Metareview:**

In this paper, the authors explore a new attack setting that would require taking adversarial latent perturbations into account, that is, when exchanging information when training distributed DNNs. To investigate the adversarial robustness in the latent space, the authors conduct an information-theoretical analysis leveraging the IB theory, showing that the compressed dimension helps improve model robustness, and also, arguing that the attack in the latent space is less effective than the input space, because it has more label-related information. Motivated by this theoretical analysis, they investigate the robustness of input-space and latent-space attacks under the same setting (especially the same budget), and thus verify that the input-space attacks indeed have higher success rates. The authors have added many new results to correspond to the reviewers’ concerns.

However, some key issues about this paper remain unsolved. First of all, I found that the concerns on the problem setting (raised by Reviewer an5M) make very much sense. It is widely accepted that adversarial perturbation constraints **only become necessary when humans are in the loop**. However, under distributed DNN training, humans are not certainly in the loop, and these constraints are unnecessary. Maybe there are some detectors (as the authors argue), but I am confident that the original $\ell_p$ ball (especially under the same budget size as for humans) does not meet a good criteria for determining its stealthiness. Thus, without further evidence, I hold a negative view of the practical significance of this setting.

Second, I also find the criticism on the theoretical results by Reviewer 4MyR reasonable. These theorems seem to be the most nontrivial part of this work. As Reviewer 4MyR pointed out, Finding 1 seems to be covered already by the ICML paper, and the new one does not make much difference. Finding 2 seems only a result of the information processing inequality, since the latent is more directly connected to the label than the input in the NNs. However I do not think we can directly conclude from this point alone that the latent would be more robust than the input. The strong assumptions could be a cause of this crude conclusion, which could be misleading.

Besides, Reviewer 4MyR also points out the lack of adaptive attacks that could lead to stronger latent space attacks. The authors do not yet provide these results.

Summarizing these points, although this paper does have good merits in its thorough empirical study, I do find these fundamental concerns on its setting and theory largely remain and dominate the essential quality of this work. Therefore, I would recommend rejection. I strongly encourage the authors to revise their papers to fix the problems above: find the right setting to compare the two attacks, and perform a more comprehensive and practical theoretical analysis that fits their real-world behaviors.

**Justification For Why Not Higher Score:**

There are some foundational concerns about the problem setup and the theoretical implications as detailed above.

**Justification For Why Not Lower Score:**

N/A

---

### Decision · Program_Chairs · 2024-01-16

Reject